 # FOCUS: Unified Vision-Language Modeling for Interactive Editing Driven by Referential Segmentation

**Fan Yang**[1,2,3] **Yousong Zhu**[4†] **, Xin Li**[2] **, Yufei Zhan**[1,3] **, Hongyin Zhao**[1] **,**
**Shurong Zheng**[1,2,3] **, Yaowei Wang**[2] **, Ming Tang**[1,3] **, Jinqiao Wang**[1,2,3,5†] **,**

[1]Foundation Model Research Center, Institute of Automation,
Chinese Academy of Sciences, Haidian District, Beijing, China
[2]Peng Cheng Laboratory, Shenzhen, China
[3]School of Artificial Intelligence, University of Chinese Academy of Science, Beijing, China
[4]School of Artificial Intelligence, China University of Mining and Technology-Beijing, Beijing, China
[5]Wuhan AI Research, Wuhan, China
{yangfan_2022, zhanyufei2021, zhengshurong2023, zhaohongyin2020}@ia.ac.cn
{tangm, jqwang}@nlpr.ia.ac.cn, zhuyousong@cumtb.edu.cn, {lix07, wangyw}@pcl.ac.cn

## Abstract

Recent Large Vision Language Models (LVLMs) demonstrate promising capabilities in unifying visual understanding and generative modeling, enabling both accurate content understanding and flexible editing. However, current approaches treat ***"what to see"*** and ***"how to edit"*** separately: they either perform isolated object segmentation or utilize segmentation masks merely as conditional prompts for local edit generation tasks, often relying on multiple disjointed models. To bridge these gaps, we introduce FOCUS, a unified LVLM that integrates segmentation-aware perception and controllable object-centric generation within an end-to-end framework. FOCUS employs a dual-branch visual encoder to simultaneously capture global semantic context and fine-grained spatial details. In addition, we leverage a MoVQGAN-based visual tokenizer to produce discrete visual tokens that enhance generation quality. To enable accurate and controllable image editing, we propose a progressive multi-stage training pipeline, where segmentation masks are jointly optimized and used as spatial condition prompts to guide the diffusion decoder. This strategy aligns visual encoding, segmentation, and generation modules, effectively bridging segmentation-aware perception with fine-grained visual synthesis. Extensive experiments across three core tasks, including multimodal understanding, referring segmentation accuracy, and controllable image generation, demonstrate that FOCUS achieves strong performance by jointly optimizing visual perception and generative capabilities.

## 1 Introduction

Large Vision-Language Models (LVLMs) are becoming a transformative paradigm in artificial intelligence [45, 2, 38]. Through large-scale pretraining, they unify visual and textual modalities and have achieved remarkable progress across a variety of tasks. These models can jointly process

---

[*]is the corresponding authors

39th Conference on Neural Information Processing Systems (NeurIPS 2025).

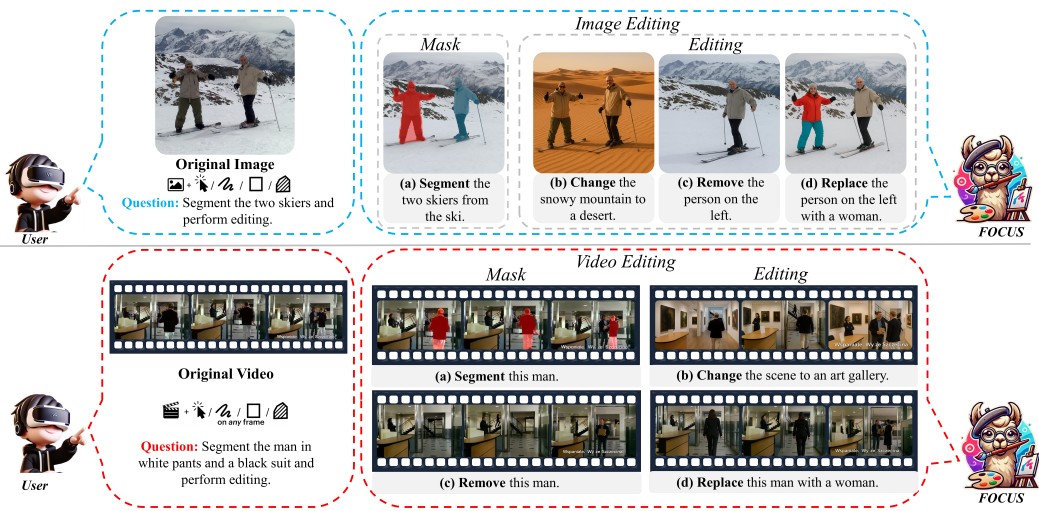

Figure 1: **FOCUS** enables fine-grained segmentation and editing for both **images** and **videos** through multimodal user instructions. It supports various region specification formats including **clicks**, **scribbles**, **boxes**, and **masks**, and for videos, **annotations on any single frame** suffice to guide full-clip editing. FOCUS can perform detailed region segmentation (e.g., identifying individual people), and supports diverse editing operations such as **removal, replacement**, and **scene transformation** across spatial and temporal domains.

images, videos, and natural language within a single architecture, demonstrating strong performance in visual question answering, image captioning, referring segmentation, and conditional generation [33, 75, 50, 38, 61]. The effectiveness of LVLMs largely stems from their ability to align multimodal semantics and generalize across diverse tasks with minimal task-specific adaptation.

At the same time, image and video generation technologies [47, 7, 46, 85] have advanced rapidly, enabling the synthesis of high-quality visual content and reshaping how we approach artistic creation and visual communication. These developments have driven increasing demands for controllability and fidelity across domains such as art, industry, and education [19]. To meet these demands, some approaches [60, 15, 10] as is shown in fig. 2(a) attempt to combine image generation models with segmentation decoders and text processing modules through modular design. However, such methods often rely on manually engineered pipelines and lack unified modeling and deep feature-level interaction. To address these limitations, others method [30] as in fig. 2(b) leverage the instruction understanding capabilities of LVLMs to dynamically dispatch pretrained expert models through task routing, enabling flexible multi-task adaptation. While these approaches improve tool orchestration, they lack deep cross-modal fusion and joint feature optimization, making it difficult to unify perception and generation effectively. More recent methods[55, 56] as is show in fig. 2(c) have begun to construct unified vision-language frameworks that combine image understanding and generation, using the generalization capability of LVLMs to bridge the semantic gap between high-level understanding and low-level synthesis. However, most of these methods still operate at a coarse level of text-driven control and struggle to support fine-grained editing or object-level manipulation.

To overcome these challenges, we propose FOCUS, an end-to-end LVLM framework that unifies segmentation-aware perception and region-controllable generation under natural language guidance, as is shown in fig. 1. The core of FOCUS features a dual-branch visual encoder, where a CLIP-like or QwenViT-style encoder extracts global semantic representations, while a hierarchical encoder based on ConvNeXt-L focuses on fine-grained local perception. This structure provides stable multi-scale segmentation support and improves adaptability to varying image resolutions. To improve visual generation, FOCUS adopts a VQGAN-based visual tokenizer, inspired by [62, 57, 21], which separately models semantic concepts and texture information. To mitigate the inevitable information loss during quantization, we retain the continuous pre-quantization features from the tokenizer as visual inputs to the language model, enabling fine-grained multimodal understanding. FOCUS is trained through a progressive multi-stage strategy, gradually increasing the input and output resolutions from low to high to ensure stable convergence and final performance. Segmentation

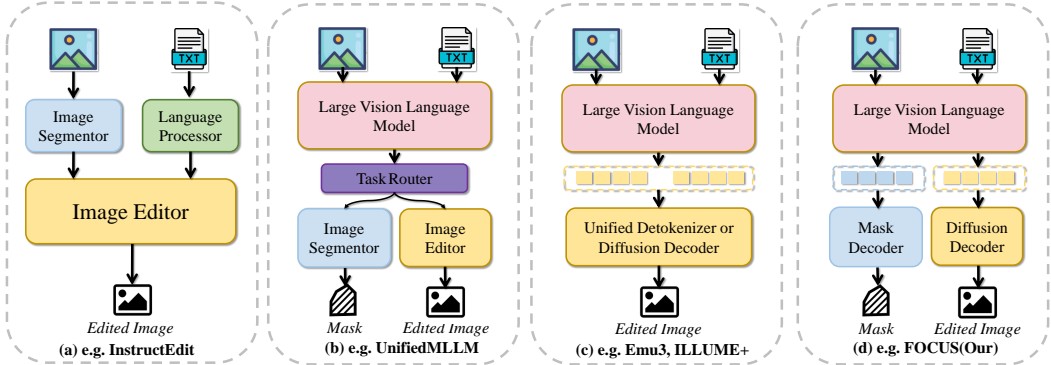

Figure 2: Comparison of controllable image editing paradigms. (a) Modular methods rely on separately trained components. (b) Task-routing LLMs orchestrate existing tools without joint modeling. (c) Unified models combine perception and generation but lack fine-grained control. (d) Our FOCUS jointly models segmentation and generation for precise, region-level editing.

masks are jointly optimized and used as spatial condition prompts to guide a diffusion-based generator for pixel-level editing. During both the multimodal pretraining and instruction tuning stages, we introduce diverse and increasingly complex task distributions, with carefully designed instruction formats, to fully enhance the model's perception understanding and generation capabilities.

Experimental results show that FOCUS achieves significant improvements across three core tasks: multimodal understanding, controllable image generation and editing, and referring segmentation. By jointly modeling pixel-level perception and generation, FOCUS demonstrates higher semantic precision and stronger spatial controllability. Further analysis reveals that pixel-level perception plays a crucial role in bridging the gap between high-level understanding and low-level synthesis. These findings confirm the feasibility and effectiveness of unifying segmentation-aware perception and controllable generation within a single LVLM framework, paving the way for interactive, precise, and generalizable multimodal editing systems.

- We propose FOCUS, a unified large vision language model that integrates pixel-level perception with region-controllable image generation and editing within an end-to-end framework.
- We develop a progressive multi-stage training pipeline that gradually increases image resolution and task complexity. This pipeline enables effective alignment and interaction between the segmentation decoder and the generation module across different scales and modalities.
- Extensive experiments on multimodal understanding, referential segmentation, and controllable image editing demonstrate that FOCUS consistently outperforms existing state-of-the-art models in both visual understanding and generation quality.

## 2 Related work

The landscape of large vision–language models (LVMs) has rapidly evolved, aiming to unify multimodal understanding and generation across images, videos, and texts. A representative example is the Emu series, which introduces autoregressive modeling to predict the next visual or textual token, thus enabling a generalist interface for diverse multimodal tasks such as image captioning, video event understanding, and cross-modal generation. This contrasts with earlier LVM designs (e.g., BLIP [27], Flamingo [1]) that bridge frozen vision and language models using separate modality connectors, often focusing solely on text prediction and neglecting direct supervision over visual signals. Emu2 [56] further extends this paradigm by positioning itself as an in-context learner, exploiting interleaved video–text data to deliver fine-grained temporal and causal reasoning over multimodal inputs.

While many of these works achieve remarkable integration of multimodal comprehension and generation, they often fall short in interactive and controllable editing. For instance, the LLaVA-Interactive [11] system explores multi-turn, multimodal interactions combining visual chat, seg-

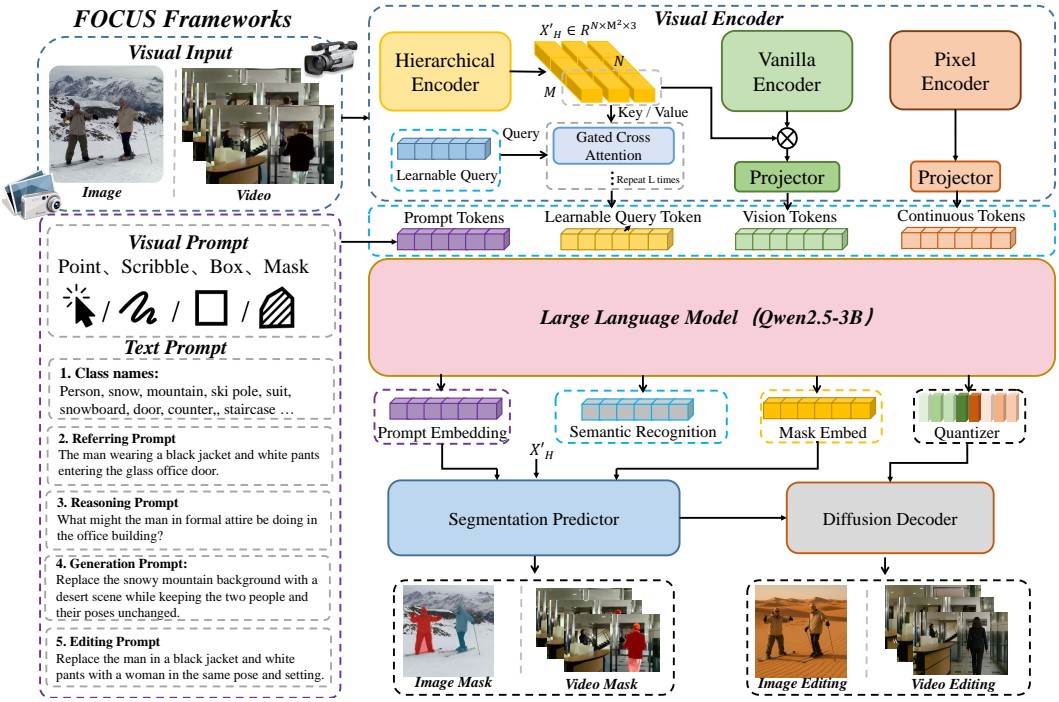

Figure 3: Overview of the FOCUS framework and training pipeline. The left part shows the unified architecture, which integrates a vanilla encoder, a hierarchical encoder, a pixel encoder, a large language model, a segmentation predictor, and a diffusion decoder for fine-grained perception and controllable image generation.

mentation, and editing but largely depends on the synergy of pretrained components without true end-to-end learning. More critically, recent editing frameworks such as InstructEdit emphasize the importance of high-quality segmentation masks by leveraging external Grounded-SAM masks to guide diffusion-based image editing. However, these approaches are not fully end-to-end; they separate mask generation from the editing pipeline, highlighting a major gap in unifying segmentation awareness with controllable generation.

In parallel, dataset-centric efforts like Localized Narratives [48] and Video Localized Narratives [59] have emerged, providing rich multimodal annotations by aligning every word or phrase with specific image or video regions. These resources enable granular referential grounding and enhance the evaluation and training of models that need to capture both spatial and temporal semantics across modalities. Additionally, Describe Anything [31] offers detailed localized captioning, helping bridge the gap between referential understanding and region-specific generation in both image and video domains.

Collectively, these prior works lay the foundation for advancing multimodal understanding, generation, and editing. Yet, there remains an unmet need for a unified architecture that couples segmentation-aware decoding with mask-driven diffusion editing in a fully end-to-end manner. This motivates the development of the proposed framework, which seeks to integrate referential localization, structured mask generation, and controllable editing into a seamless multimodal pipeline.

## 3  Methodology

As illustrated in Fig.3, FOCUS is a unified large vision-language model that integrates pixel-level perception and controllable image generation in an end-to-end framework. The model comprises four core components, covering the entire process from visual encoding to image synthesis. Section 3.1 introduce a dual-branch visual encoder and a generation-oriented visual tokenizer are employed to extract global semantic features and multi-scale fine-grained representations from inputs at different resolutions, discretizing the visual content into high-level tokens suitable for downstream editing tasks. Next, in the  Section 3.2 a large language model based on Qwen2.5 is introduced to unify

multimodal input formats through task prompts, enabling support for a wide range of vision-language tasks. Building on this, in the Section 3.3 a segmentation decoder is designed to perform high-precision fine-grained object segmentation. A diffusion-based image generator then synthesizes high-fidelity images guided by spatial segmentation masks and conditioned on discrete visual tokens in the Section 3.4. In the Section 3.5 further details the progressive training strategy and the construction of multi-source datasets.

## 3.1 Dual Visual Encoders and Generative Tokenizer

A fundamental challenge in building unified large vision-language models for both image understanding and generation lies in the significant discrepancy between high-level semantic understanding and low-level fine-grained image synthesis. This discrepancy often leads to mutual interference and optimization conflicts during training. To address this issue, we propose a novel architecture that combines a dual-branch visual encoder with a generative visual tokenizer. This design preserves the ability to model global high-level semantic features while enhancing the representation of fine-grained, multi-scale low-level visual details. Meanwhile, the visual tokenizer discretizes continuous visual information, effectively extending the representational capacity of the image generation module.

Specifically, we process each image at two resolutions. A low-resolution image $X_L \in \mathbb{R}^{3 \times 256 \times 256}$ is fed into a **Vanilla Encoder** (based on QwenViT [3]) to capture global semantic information. Simultaneously, a high-resolution image $X_H \in \mathbb{R}^{3 \times 768 \times 768}$ is encoded by a **Hierarchical Encoder** (ConvNeXt-L [41]) to obtain high-resolution, multi-scale visual features. These two branches are fused using a cross-attention mechanism to enhance low-resolution features with local detail:

$$X'_H = \text{ConvNeXt}(X_H), \quad X'_L = \text{QwenViT}(X_L) \tag{1}$$

$$E'_{\text{img}} = \text{CrossAttn}(X'_L, X'_H), \quad E_{\text{img}} = \text{MLP}(E'_{\text{img}}) + E'_{\text{img}} \tag{2}$$

To further inject fine-grained visual information into the language model, we introduce a gated cross-attention adapter that enhances learnable queries with high-resolution visual features at multiple scales. For the $l$-th layer query $h^{(l)}$, and $j$-th scale feature $f_{\text{img}}^{(j)}$, the fusion is formulated as:

$$h^{(l)'} = h^{(l)} + \tanh(\gamma^{(l)}) \cdot \text{CrossAttn}(h^{(l)}, G_p(f_{\text{img}}^{(j)})) \tag{3}$$

$$h_{\text{Adapter}}^{(l)} = h^{(l)'} + \tanh(\beta^{(l)}) \cdot \text{FFN}(h^{(l)'}) \tag{4}$$

where $G_p(\cdot)$ is a projection function aligning the visual feature space, and $\gamma^{(l)}, \beta^{(l)}$ are learnable scaling parameters initialized to zero. This mechanism injects spatial detail while maintaining the efficiency of learnable queries, enhancing both segmentation and generative capabilities.

To enhance the upper bound of image generation quality, we introduce a **generative visual tokenizer** based on the MoVQGAN [84] architecture to discretize visual information. The visual tokenizer serves as a key component for unified autoregressive image generation, but its quantization process inevitably introduces information loss. To address this, we use the continuous features before quantization as the visual input to the large language model (LLM), which provides a more informative representation for fine-grained multimodal understanding. The LLM then outputs discrete visual tokens that guide the diffusion model for high-quality image synthesis.

Specifically, we use the fused dual-branch features $E'_{\text{img}}$ as the global semantic representation for generation, which are quantized into discrete semantic tokens and decoded through a lightweight decoder. During the visual tokenizer's pretraining stage, this semantic branch is supervised with a cosine similarity loss against the original encoder features. A downsampling rate of $28\times$ is adopted to align with mainstream vision-language models and focus on high-level semantic concepts.

We further extend the generative visual tokenizer with a **pixel branch**, following the standard MoVQGAN design. This branch uses a $16\times$ downsampling rate to preserve fine textures. After quantization, the semantic and pixel tokens are concatenated along the channel dimension and passed to the decoder for image reconstruction. The pixel branch is trained with a combination of L1 loss, perceptual loss, and adversarial loss. We use large codebooks for both branches, with a size of 32,768 for the semantic branch and 98,304 for the pixel branch.

Together, the generative visual tokenizer and dual-branch visual encoder construct a unified and expressive visual representation that enables **FOCUS** to support both fine-grained perception and controllable image generation within a single framework.

## 3.2 Large Vision-Language Model and Input Schema

In **FOCUS**, we adopt **Qwen2.5-3B** [71] as the large language model (LLM) to model unified multimodal inputs from the visual encoder and task instructions. The model takes as input a four-tuple $(E_{\text{img}}, X_S, X_T, h_{\text{Adapter}}^{(l)})$, where $E_{\text{img}}$ denotes the fused visual features from the dual-branch encoder, $X_S$ represents the structured textual prompts, $X_T$ refers to the continuous pixel-level visual features from the pixel encoder, and $h_{\text{Adapter}}^{(l)}$ is the learnable query feature obtained through multi-scale cross-modal adaptation.

These inputs are fed into the LLM to produce the output representation:

$$E_O = F_{\text{LLM}}(E_{\text{img}}, X_S, X_T, h_{\text{Adapter}}^{(l)})$$

From the LLM output $E_O$, we extract two key components. First, we introduce a decoding constraint that enforces the model to predict the names of all present objects before generating the mask tokens. This prior step encourages the generation of semantically enriched mask tokens, which incorporate global semantic context from the image and are subsequently used by the segmentation module to produce masks.

In addition, the LLM supports autoregressive prediction of discrete visual tokens. Following a semantic-first strategy, the model first generates semantic tokens to define global content structure, then generates pixel tokens to recover fine visual textures. This two-stage decoding improves the alignment between textual instructions and generated visual content.

**Prompt Design.** The input prompt schema consists of two components: the *task instruction prompt* $S_I$ and the *condition prompt* $S_C$. The task instruction prompt specifies the objective of the model in natural language, guiding the LLM to perform segmentation, generation, or editing accordingly. For example, in class-based segmentation tasks such as panoptic, open-vocabulary, or video instance segmentation, the instruction can be *"Please segment all the positive objects by the following candidate categories."* In referring or reasoning segmentation tasks (e.g., RES [23, 76, 44, 34], R-VOS [28], ReasonVOS [70]), the instruction becomes *"Please segment the target referred to by the language description."* For visual-guided tasks such as interactive or video object segmentation, the instruction is phrased as "Please segment according to the given visual reference regions."

The condition prompt provides additional information specific to each task. In class-based segmentation, it lists the candidate category labels; in referring segmentation, it provides a natural language expression; in visual-guided settings, it includes pooled region features sampled from CLIP embeddings at specified coordinates. For image generation tasks, the condition prompt consists of a text description such as *"a cat sitting on a couch,"* while for image editing tasks, it contains language-based or spatial referring expressions like *"replace the person on the right with a dog."* The condition prompt not only supplies contextual information but also serves as an implicit classifier for category-aware mask prediction.

## 3.3 Segmentation Module

The segmentation predictor $F_p$ takes three types of inputs: task-specific prompt embeddings $\{E_P^k\}_{k=1}^K$, semantic-enhanced mask token embeddings $\{E_Q^j\}_{j=1}^N$ from the LLM output, and multi-scale visual features $f_{\text{img}} = X_H'$ from the visual encoder. Here, $K$ denotes the number of candidate categories, and $N$ represents the number of mask proposals. These inputs are fused to predict segmentation outputs in the following form:

$$\{(m_j, z_j, e_j)\}_{j=1}^N = F_p\left(\{E_P^k\}_{k=1}^K, \{E_Q^j\}_{j=1}^N, f_{\text{img}}\right), \tag{5}$$

where $m_j \in \mathbb{R}^{H \times W}$ denotes the $j$-th predicted binary mask, $z_j \in \mathbb{R}^K$ is the associated category score vector, and $e_j \in \mathbb{R}^D$ is an optional instance-level embedding produced by an auxiliary embedding head to enable temporal association in video segmentation tasks.

For video tasks, we adopt a frame-by-frame processing strategy to generate frame-level segmentation results, ensuring efficient training and inference while maintaining temporal consistency.

To support multi-scale supervision in the subsequent image generation stage, the segmentation module consistently produces fixed-resolution masks based on $X_H'$, providing stable spatial guidance. These

masks are then rescaled to match the resolution used by the diffusion model, enabling effective alignment between segmentation outputs and the image synthesis pipeline.

## 3.4 Diffusion Decoder for Controllable Generation

To enable high-quality and region-controllable image generation, FOCUS incorporates a latent diffusion decoder initialized from SDXL. The generation process is formulated as denoising over latent variables, conditioned on structured visual prompts. We utilize semantic tokens $z_{\text{sem}}$ and pixel-level tokens $z_{\text{pix}}$ generated by the large language model. These tokens are mapped into continuous embeddings via a learned codebook and concatenated with noisy latents before being passed into a UNet-based denoising backbone.

To achieve spatial control, we leverage predicted segmentation masks $m_j \in \mathbb{R}^{H \times W}$ from the segmentation module. These masks are first downsampled to match the latent spatial resolution, yielding $\tilde{m}_j \in \mathbb{R}^{H' \times W'}$, where $H' \times W'$ denotes the latent feature resolution. The downsampled mask is then flattened and passed through a linear projection layer to obtain a spatial guidance sequence $f_m \in \mathbb{R}^{L \times C}$, where $L = H' \times W'$ and $C$ is the channel dimension. This sequence is injected into the UNet through cross-attention at intermediate layers to guide regional generation. The interaction is formalized as:

$$\hat{z}_t = \text{CrossAttn}(\phi(z_t), f_m), \quad z_{t+1} = \text{UNet}(\hat{z}_t), \tag{6}$$

where $\phi(z_t)$ denotes the projected latent features at denoising timestep $t$, and $f_m$ provides the spatial condition derived from $\tilde{m}_j$.

We also experimented with using mask token embeddings $\{E_Q^j\}$ as alternative conditioning inputs, but found that direct spatial control through explicit masks yields better localization and more faithful region editing in our experiments.

## 3.5 Training Procedure and Data Composition

To support unified pixel-level understanding, high-level multimodal reasoning, and spatially controllable image generation, FOCUS adopts a progressive four-stage training paradigm that incrementally builds visual tokenization, vision-language alignment, and conditional generation capabilities.

**Stage 1: Pretraining of Dual-Branch Visual Tokenizer and Diffusion Decoder.** We first train a semantic branch and a pixel branch to discretize input images into semantic and texture tokens using SimVQ quantizers. A progressive resolution strategy is adopted, starting from $256 \times 256$ and gradually increasing to $512 \times 512$. A large-scale image-text dataset of approximately 58M pairs is constructed based on resolution constraints and visual diversity. On top of this, we pretrain a latent diffusion decoder using a 10M image subset, enabling high-fidelity reconstruction from discrete tokens. This stage focuses purely on image modeling and does not involve segmentation or controllable generation.

**Stage 2: Visual-Language Adapter Warmup.** To align visual features with the input space of the large language model (LLM), we train projection heads for both the vanilla encoder and the pixel encoder, along with learnable queries and gated cross-attention modules in the hierarchical encoder branch. All visual backbone encoders are kept frozen. Training is conducted at $256 \times 256$ resolution with standard language modeling loss.

**Stage 3: Multimodal Pretraining with Segmentation-Aware Alignment.** In this stage, we jointly train the LLM, visual adapters, and the mask decoder to improve the model's perception of segmentation structures and enhance multimodal generation capability. Training is performed in two phases with input resolutions of $256 \times 256$ and $512 \times 512$. Supervision includes token-level cross-entropy loss, segmentation mask supervision (Dice + BCE), and image reconstruction loss (L2) from diffusion outputs. The training corpus includes multimodal data spanning image-text pairs, segmentation tasks, and vision-language reasoning. Full dataset details are presented in appendix.

**Stage 4: Instruction Tuning for Region-Controlled Editing.** This stage introduces region-level controllability for editing and generation tasks. We jointly train the LLM, the mask decoder, and

cross-attention layers within the diffusion model, while freezing the visual encoders, visual tokenizer, and VAE components. Input images are resized using a bucketed aspect ratio cropping strategy with total pixel counts ranging from $512^2$ to $1024^2$. Supervision includes token prediction (cross-entropy), segmentation mask prediction (Dice + BCE), and image reconstruction (L2 loss). Detailed task and dataset configurations are summarized in appendix.

# 4 Experiments

We conduct comprehensive experiments to evaluate the performance of FOCUS across three representative tasks: multimodal understanding, referential segmentation, and generation and controllable editing. All experiments are designed to validate the model's ability to unify perception and generation in an end-to-end framework, with strong alignment to natural language instructions, fine-grained localization, and object-aware editing fidelity.

## 4.1 Implementation Details

In our experiments, we adopt Qwen2.5 as the backbone large language model (LLM). The visual encoder consists of two branches: the semantic decoder is composed of four attention blocks with 2D relative positional encoding (2D-RoPE), while the pixel encoder and decoder follow a MoVQGAN-based architecture with base channel dimensions of 128 and 384, respectively. The codebook size is set to 32,768 for the semantic branch and 98,304 for the pixel branch, with both using a code dimension of 32. We employ the AdamW optimizer without weight decay and use a constant learning rate across the visual encoder, diffusion decoder, and the large vision language model. Detailed training hyperparameters for each component are summarized in the supplementary materials. The training of the Dual-Branch Visual Tokenizer and the diffusion decoder each took approximately 3 days on a computing cluster, while the 3B-parameter MLLM required around 13 days to complete the three-stage training process.

Table 1: FOCUS performance on general and document-oriented benchmarks.

| Method | LLM | General | | | | | | | Doc | | | | |
| | | POPE | MMBench | SEED | MME-P | MM-Vet | MMMU | AI2D | VQA-text | ChartQA | DocVQA | InfoVQA | OCRBench |
| --- | --- | --- | --- | --- | --- | --- | --- | --- | --- | --- | --- | --- | --- |
| Understanding Only | | | | | | | | | | | | | |
| InstructBLIP [16] | Vicuna-7B [33] | - | 36.0 | 53.4 | - | 26.2 | 30.6 | 33.8 | 50.1 | 12.5 | 13.9 | - | 276 |
| Qwen-VL-Chat [3] | Qwen-7B | - | 60.6 | 58.2 | 1487.5 | - | 35.9 | 45.9 | 61.5 | 66.3 | 62.6 | - | 488 |
| LLaVA-1.5 [38] | Vicuna-7B | 85.9 | 64.3 | 58.6 | 1510.7 | 31.1 | 35.4 | 54.8 | 58.2 | 18.2 | 28.1 | 25.8 | 318 |
| ShareGPT4V [9] | Vicuna-7B | - | 68.8 | 69.7 | 1567.4 | 37.6 | 37.2 | 58 | 60.4 | 21.3 | - | - | 371 |
| LLaVA-NeXT [25] | Vicuna-7B | 86.5 | 67.4 | 64.7 | 1519 | 43.9 | 35.1 | 66.6 | 64.9 | 54.8 | 74.4 | 37.1 | 532 |
| Emu3-Chat [62] | 8B from scratch | 85.2 | 58.2 | 68.2 | - | 37.2 | 31.6 | 70.0 | 64.7 | 68.6 | 76.3 | 43.8 | 687 |
| Unify Understanding and Generation | | | | | | | | | | | | | |
| Unified-IO [43] | 6.8B from scratch | 87.7 | - | 61.8 | - | - | - | - | - | - | - | - | - |
| Chameleon [57] | 7B from scratch | - | - | - | - | 8.3 | 22.4 | - | - | - | - | - | - |
| LWM [28] | LLaVA-2-7B | 75.2 | - | - | - | 9.6 | - | - | 18.8 | - | - | - | - |
| Show-o [68] | Phi-1.5B | 73.8 | - | - | 948.4 | - | 25.1 | - | - | - | - | - | - |
| VILA-U (256) [67] | LLaMA-2-7B | 83.9 | - | 56.3 | 1336.2 | 27.7 | - | - | 48.3 | - | - | - | - |
| VILA-U (384) [67] | LLaMA-2-7B | 85.8 | - | 59 | 1401.8 | 33.5 | - | - | 60.8 | - | - | - | - |
| Janus [66] | DeepSeek-LLM-1.3B | 87.0 | 69.4 | 63.7 | 1338.0 | 34.3 | 30.5 | - | - | - | - | - | - |
| Janus-Pro-1B [13] | DeepSeek-LLM-1.3B | 86.2 | 75.5 | 68.3 | 1444.0 | 39.8 | 36.3 | - | - | - | - | - | - |
| Janus-Pro-7B [13] | DeepSeek-LLM-7B | 87.4 | 79.2 | 72.1 | 1567.1 | 50.0 | 41.0 | - | - | - | - | - | - |
| ILLUME+ [21] | Qwen2.5-3B | 87.6 | 80.8 | 73.3 | 1414.0 | 40.3 | 44.3 | 74.2 | 69.9 | 69.9 | 80.8 | 44.1 | 672 |
| FOCUS | Qwen2.5-3B | **88.0** | **81.5** | **73.9** | **1570.3** | **50.2** | **44.9** | **74.5** | **70.4** | **70.3** | **81.1** | **44.3** | _678_ |

## 4.2 Multimodal understanding

To evaluate the multimodal understanding capabilities of our model, we conduct systematic evaluations on two categories of widely-used benchmarks, as is show in table 1: (1) General benchmarks, including POPE, MMBench, SEED, MME-P, MM-Vet, MMMU, and AI2D; and (2) Document-oriented benchmarks, including VQA-text, ChartQA, DocVQA, InfoVQA, and OCRBench. Experimental results show that, despite using only a 3B-parameter model, FOCUS achieves performance comparable to state-of-the-art unified models such as Janus-Pro-7B and ILLUME-7B, and notably outperforms ILLUME+ with the same parameter scale. This performance gain is largely attributed to the incorporation of multi-scale high-resolution features and segmentation masks, which significantly enhance pixel-level perception.

## 4.3 Controllable Generation and Editing

Multimodal image generation. To evaluate the multimodal visual generation capability, we use the MJHQ-30K, GenAI-bench and GenEval benchmarks in the table 2. For MJHQ-30K, we adopt the

Fréchet Inception Distance (FID) metric on 30K generated images compared to 30K high-quality images, measuring the generation quality and diversity. GenAI-bench and GenEval are challenging text-to-image generation benchmarks designed to reflect the consistency between text descriptions and generated images. We compare FOCUS with previous state-ofthe-art multimodal generation-only and unified models. This highlights the superior generation quality and diversity enabled by our diffusion-based approach. Additionally, FOCUS achieves competitive results on the GenAI-bench and GenEval benchmarks and attains the highest accuracy in advanced categories on GenAI-bench, demonstrating its ability to understand and generate images from complex text descriptions. Figure 7 shows more results of FOCUS on generating flexible resolution images.

Table 2: **Evaluation results on multimodal image generation benchmarks.**

| Method | Params. | Type | MJHQ30K FID | GenAI-bench Basic | GenAI-bench Adv. | GenEval Overall | Single Obj | Two Obj. | Counting | Colors | Position | Color Attri. |
|---|---|---|---|---|---|---|---|---|---|---|---|---|
| *Generation Only* | | | | | | | | | | | | |
| SDv1.5 [51] | 0.9B | Diffusion | - | - | - | 0.43 | 0.97 | 0.38 | 0.35 | 0.76 | 0.04 | 0.06 |
| PixArt-α [7] | 0.6B | Diffusion | 6.14 | - | - | 0.48 | 0.98 | 0.45 | 0.44 | 0.08 | 0.07 | 0.07 |
| SDXL [47] | 2.6B | Diffusion | 9.55 | **0.83** | 0.63 | 0.55 | 0.98 | 0.41 | 0.48 | 0.15 | 0.17 | 0.23 |
| Emu3-Gen [62] | 8B | Autoreg. | - | - | - | 0.54 | 0.98 | 0.71 | 0.34 | 0.81 | 0.17 | 0.21 |
| *Unify Understanding and Generation* | | | | | | | | | | | | |
| Chameleon [57] | 7B | Autoreg. | - | - | - | 0.39 | - | - | - | - | - | - |
| LWM [35] | 7B | Autoreg. | 17.77 | 0.63 | 0.53 | 0.47 | 0.93 | 0.41 | 0.46 | 0.79 | 0.09 | 0.15 |
| Show-o [68] | 1.5B | Autoreg. | 15.18 | 0.70 | 0.60 | 0.45 | 0.95 | 0.52 | 0.49 | 0.83 | 0.28 | 0.30 |
| VILA-U(256) [67] | 7B | Autoreg. | 12.81 | 0.72 | 0.64 | - | - | - | - | - | - | - |
| VILA-U(384) [67] | 7B | Autoreg. | 7.69 | 0.71 | 0.66 | - | - | - | - | - | - | - |
| Janus [66] | 7B | Autoreg. | 10.1 | - | - | - | - | - | - | - | - | - |
| Janus-Pro-1B [13] | 1.3B | Autoreg. | - | 0.73 | 0.68 | 0.61 | 0.99 | 0.82 | 0.51 | 0.86 | 0.39 | 0.26 |
| Janus-Pro-7B [13] | 7B | Autoreg. | - | 0.80 | 0.69 | 0.59 | **0.90** | 0.59 | **0.90** | 0.79 | 0.66 | 0.25 |
| ILLUME+ [21] | 3B | Autoreg. | 6.00 | 0.72 | 0.71 | 0.72 | 1.00 | **0.99** | 0.88 | 0.62 | **0.84** | 0.53 |
| FOCUS | 3B | Autoreg. | **6.05** | **0.83** | **0.72** | **0.75** | **1.20** | 0.98 | 0.87 | 0.66 | 0.81 | **0.57** |

Multimodal image editing. To assess the multimodal image editing capability of our method, we evaluate it on the Emu Edit benchmark and report the CLIP-I, CLIP-T, CLIP-DIR and DINO scores. The CLIP-I and DINO scores measure the model's ability to preserve elements from the source image, while the CLIP-T and CLIP-DIR score measures the consistency between the output image and the target caption. As illustrated in the table 4, our model demonstrates strong performance in image editing tasks, surpassing specialized models, particularly in the CLIP-T metric. This indicates that the unified model's superior understanding enhances its ability to interpret editing instructions, resulting in more precise modifications.

Table 3: **Comparisons with other referring segmentation.**

| Method | RefCOCO Val | testA | testB | Refcoco+ Val | testA | testB | Refcocog Val | Test | gRefCOCO Val | testA | testB |
|---|---|---|---|---|---|---|---|---|---|---|---|
| *Segmentation Specialist* | | | | | | | | | | | |
| CRIS [80] | 70.5 | 73.2 | 66.1 | 62.3 | 68.1 | 53.7 | 59.9 | 60.4 | 55.3 | 63.8 | 51.0 |
| LAVT [74] | 72.7 | 75.8 | 68.8 | 62.1 | 68.4 | 55.1 | 61.2 | 62.1 | 57.6 | 65.3 | 55.0 |
| PolyFormer-B [39] | 74.8 | 76.6 | 71.1 | 67.6 | 72.9 | 58.3 | 67.8 | 69.1 | - | - | - |
| *LVLM-based Segmentation Network* | | | | | | | | | | | |
| LISA-7B [24] | 74.9 | 79.1 | 72.3 | 65.1 | 70.9 | 58.1 | 67.9 | 70.6 | 38.7 | 52.6 | 44.8 |
| PixelLM7B [50] | 73.0 | 76.5 | 68.2 | 66.3 | 71.7 | 58.3 | 69.3 | 70.5 | - | - | - |
| PSALM [82] | 83.6 | 84.7 | 81.6 | 72.9 | 75.5 | 70.1 | 73.8 | 74.4 | 42.0 | 52.4 | 50.6 |
| HyperSeg [64] | **84.8** | 85.7 | **83.4** | **79.0** | 83.5 | **75.2** | **79.4** | 78.9 | 47.5 | 57.3 | 52.5 |
| FOCUS | 84.1 | **86.3** | 82.7 | 78.5 | **84.1** | 74.3 | 79.3 | **79.8** | 48.7 | 58.5 | 53.0 |

## 4.4 Referring Segmentation Accuracy

Table 4: **Quantitative results on image editing benchmarks.** The performance with top-1 and top-2 value are denoted in bold and underline.

| Method | Type | Tasks | Emu Edit [53] DINO | CLIP-I | CLIP-T | CLIP-DIR |
|---|---|---|---|---|---|---|
| InstructPix2Pix [1] | Diffusion | Edit only | 0.762 | 0.834 | 0.219 | 0.078 |
| MagicBrush [81] | Diffusion | Edit only | 0.776 | 0.838 | 0.222 | 0.09 |
| OmniGen [55] | Diffusion | Edit only | 0.804 | 0.836 | 0.233 | - |
| Emu Edit [53] | Diffusion | Edit only | 0.819 | 0.859 | 0.231 | **0.109** |
| PUMA [18] | AR | Edit only | 0.785 | 0.846 | 0.270 | - |
| ILLUME | AR | Und, Gen, Edit | 0.791 | **0.879** | 0.260 | - |
| ILLUME+ | AR | Und, Gen, Edit | 0.826 | 0.872 | 0.275 | 0.101 |
| FOCUS | AR | Und, Gen, Edit | **0.831** | 0.876 | **0.278** | 0.105 |

We evaluate the referential segmentation performance of **FOCUS** on four standard benchmarks: RefCOCO, RefCOCO+, RefCOCOg, and gRefCOCO, using mean Intersection-over-Union (mIoU) as the evaluation metric. As shown in table 3, **FOCUS** achieves competitive or superior performance compared to both segmentation-specific and LVLM-based methods. These results demonstrate the effectiveness of **FOCUS** in pixel-level target localization and its strong capacity to align complex referring expressions with visual semantics in an end-to-end framework. More empirical evidence is presented in the appendix.

## 4.5 Ablation Studies

### 4.5.1 The effect of multi-stage training

The following table shows the results of using different image resolutions (256, 512, and 1024) for all-stage task training. We observe a clear trend: higher image resolutions improve the model's performance in image generation and segmentation, but lead to a decline in language understanding.

Table 5: The effect of different image resolutions and multi-stage training strategy on model performance across various tasks. ↑ indicates higher is better, ↓ indicates lower is better.

| Image Size | MJHQ30K | GenAI-Bench | | Image Understanding | | | RefCOCO | | |
|---|---|---|---|---|---|---|---|---|---|
| | FID(↓) | Basic(↑) | Adv.(↑) | POPE(↑) | MMB(↑) | SEED(↑) | testA(↑) | TestB(↑) | Valid(↑) |
| 256 | 11.85 | 0.63 | 0.56 | 85.3 | 73.1 | 70.1 | 78.5 | 79.2 | 75.4 |
| 512 | 7.39 | 0.70 | 0.61 | 87.3 | 80.6 | 71.3 | 81.3 | 82.4 | 79.8 |
| 1024 | 6.03 | 0.76 | 0.69 | 77.6(↓) | 68.2(↓) | 68.8(↓) | 83.4 | 84.9 | 81.5 |
| 256->1024(Our) | 6.05 | 0.83 | 0.72 | 88.0 | 81.5 | 73.9 | 84.1 | 86.3 | 82.7 |

Our analysis suggests that this decline is mainly due to an increase in hallucination issues at higher resolutions—when processing high-resolution images, the model is more likely to generate text that is inconsistent with the visual input. This increased hallucination negatively impacts the model's language comprehension, highlighting the challenge of balancing visual precision and robust language understanding in multimodal models.

To address this, our multi-stage training strategy progressively increases image resolution and task complexity, starting from simpler tasks at lower resolutions. This approach enables strong and balanced performance across all tasks.

### 4.5.2 The effect of multi-scale features in Gated Cross-Attention

Table 6: Ablation study on the effect of multi-scale feature fusion in Gated Cross-Attention.

| Model Variant | Scales | RefCOCO mIoU(↑) | MMBench Acc.(↑) | CLIP-T(Edit)(↑) |
|---|---|---|---|---|
| FOCUS-SingleScale (Fine) | 1 | 79.5 | 72.8 | 0.265 |
| FOCUS-SingleScale (Coarse) | 1 | 78.2 | 73.9 | 0.260 |
| FOCUS-DualScale | 2 | 80.5 | 73.5 | 0.270 |
| FOCUS (Ours) | 3 | 81.4 | 73.9 | 0.275 |

To effectively fuse multi-scale features, our model employs a Gated Cross-Attention module to process outputs from the ConvNeXt-L backbone, achieving a balance between fine-grained detail and global contextual understanding. Specifically, we design a 3-layer Gated Cross-Attention adapter that hierarchically integrates three distinct feature scales: the first layer focuses on fine features ($f_2$), the second on mid-level features ($f_3$), and the third on coarse semantic features ($f_4$).

As illustrated in the table, we compare our three-scale model ($f_2$, $f_3$, $f_4$) with variants that utilize only a single scale ($f_4$), dual scales ($f_3$, $f_4$), and all four scales ($f_2$-$f_4$). The results (mIoU on RefCOCO) demonstrate that our multi-scale fusion strategy is essential for accurately segmenting fine details, providing the optimal balance between performance and computational efficiency.

### 4.5.3 The effect of continuous visual tokens

Table 7: Ablation study on the effect of continuous visual token input on model performance.

| Continuous Input | MJHQ30K | GenAI-Bench | | Image Understanding | | |
|---|---|---|---|---|---|---|
| | FID↓ | Basic↑ | Adv.↑ | POPE↑ | MMB↑ | SEED↑ |
| ✗ | 6.85 | 0.68 | 0.65 | 82.1 | 50.4 | 56.0 |
| ✓ | 6.05 | 0.83 | 0.72 | 85.3 | 70.9 | 66.6 |

Our ablation study demonstrates that feeding continuous visual features directly into the LLM is essential for optimal performance. By bypassing discrete tokenization, this approach effectively reduces information loss and retains fine-grained visual details. As a result, the LLM receives a richer and more precise visual representation, which significantly enhances its ability to comprehend complex multimodal scenarios and accurately interpret referential instructions.

## 5 Conclusion

FOCUS demonstrates robust generalization and strong task performance across multimodal dialogue, referential segmentation, and controlled visual editing. The consistent improvements over baseline models affirm the effectiveness of our unified architecture, which tightly integrates segmentation-aware perception with instruction-guided generation. These results further support the practical value of FOCUS in real-world multimodal interaction scenarios.

# 6 Acknowledgement

This work was supported by the National Key Research and Development Program of China (No. 2024YFE0210600), the National Key R&D Program of China under Grant No. 2022ZD0160601, the National Natural Science Foundation of China under Grant Nos. 62176254 and 62276260, and the Beijing Natural Science Foundation (No. L247028).

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

# A  Dual Quantizer and Diffusion Decoder Training

To enable high-quality image generation, FOCUS pretrains dual quantizers with a three-branch visual encoder and subsequently trains a diffusion decoder conditioned on the tokens. This section details the training process for both the quantizers and the diffusion decoder.

**Dual Quantizer Training.** As illustrated in Fig. 4(a), the semantic quantizer is constructed from two visual branches: a *vanilla encoder* that captures global semantics from low-resolution inputs,

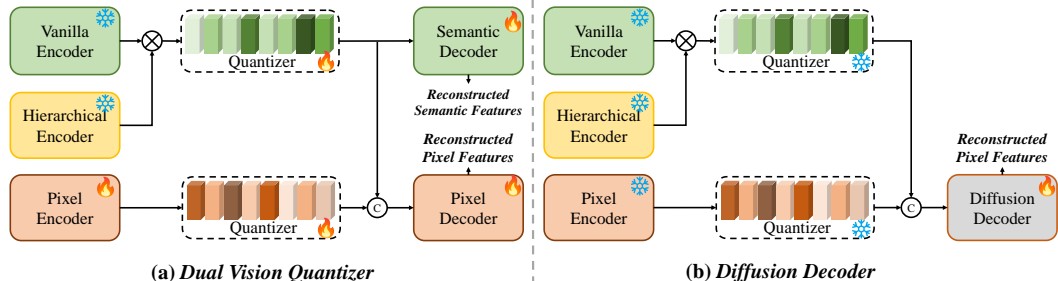

**(a)** *Dual Vision Quantizer*          **(b)** *Diffusion Decoder*

Figure 4: (a) Dual Vision Quantizer: Two separate branches encode semantic and pixel information via quantization and reconstruction. (b) Diffusion Decoder: Reuses pretrained tokens to condition UNet-based denoising for image reconstruction.

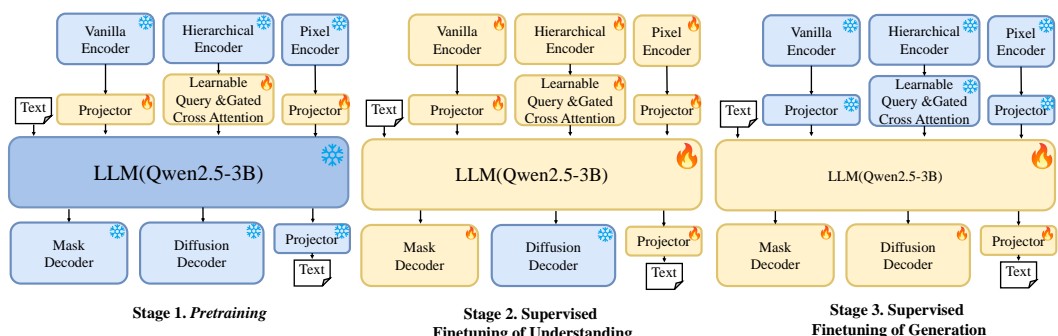

**Stage 1.** *Pretraining*          **Stage 2. Supervised Finetuning of Understanding**          **Stage 3. Supervised Finetuning of Generation**

Figure 5: Three-stage progressive training strategy of FOCUS. Frozen modules are shown in blue, trainable modules in orange. Each stage incrementally activates relevant components to align semantic perception and generative editing.

and a *hierarchical encoder* that extracts high-resolution spatial features. These two streams cross attention to provide rich semantic representations, which are quantized via a SimVQ [86] module and reconstructed using a lightweight transformer as the semantic decoder.

In parallel, a dedicated *pixel encoder* processes high-resolution images to extract fine-grained, low-level textures. Following the MoVQGAN design [84], its features are quantized and decoded via a pixel-level decoder. This branch is supervised using L1, perceptual, and adversarial losses.

Each quantizer maintains a separate codebook: 32,768 entries for semantic tokens and 98,304 entries for pixel tokens as [21]. The quantizer is trained progressively from $256 \times 256$ to $512 \times 512$ resolution using a bucketed batching strategy, and optimized on a corpus of 45M diverse image-text pairs.

**Diffusion decoder training.** After quantizer training, we train a UNet-based diffusion decoder, initialized from SDXL [47], to reconstruct high-resolution images conditioned on the learned discrete tokens. As shown in Fig. 4(b), semantic and pixel tokens are embedded into continuous representations and concatenated with noisy latents to guide the denoising process.

To accommodate diverse image shapes, 11 aspect-ratio-specific canvas sizes are predefined: {1:1, 3:4, 4:3, 2:3, 3:2, 1:2, 2:1, 1:3, 3:1, 1:4, 4:1} [21], and samples with more than 20% content loss after cropping are discarded. The training proceeds in two stages: the first at $512 \times 512$ resolution and the second at $1024 \times 1024$ for super-resolution refinement. During this stage, all encoders and codebooks are frozen, and only the diffusion decoder is updated.

# B   Progressive Large Vision Language Model Training

A core challenge in building a unified large vision language model lies in the optimization conflict between high-level semantic understanding in language and low-level visual synthesis in image generation. FOCUS addresses this by adopting a progressive three-stage training paradigm (Fig. 5)

that incrementally activates model components to align pixel-level perception and controllable image generation.

**Visual-Language Projector Warmup.** This stage establishes initial alignment between visual features and the large language model (LLM). We train only the projection heads of the vanilla and pixel encoders, along with the learnable queries and gated cross-attention modules in the hierarchical encoder. All backbone encoders, the LLM, the mask decoder is Mask2Former [14], and the diffusion decoder are frozen. Training is performed on $256\times256$ image-text pairs using a standard language modeling objective, without any segmentation or generation supervision.

**Multimodal Pretraining with Segmentation-Aware Alignment.** We activate and jointly train the LLM, visual-language adapters, and the mask decoder to enhance multimodal understanding and segmentation capability. The diffusion decoder remains frozen. Supervision includes token-level cross-entropy, segmentation losses (Dice and BCE), and visual reconstruction losses using precomputed features. Training is conducted at both $256\times256$ and $512\times512$ resolutions on datasets covering image-text alignment, referring segmentation, and multimodal reasoning.

**Instruction Tuning for Region-Controlled Editing.** This stage focuses on segmentation perception and controllable generation for LVLMs. We instruct tuning the LLM, the mask decoder, and the cross-attention layers in the diffusion decoder, while freezing all visual encoders. Images are processed at resolutions up to $1024\times1024$. Segmentation outputs are used as spatial guidance and injected into the diffusion decoder's attention layers. Supervision includes text generation loss, segmentation loss, and L2 loss between denoised outputs and targets.

Table 8: Dataset distribution across training stages in FOCUS

| Stage | Number | Task | Source |
|---|---|---|---|
| Stage I: Visual Quantizer and Diffusion Pretraining | 45M | Image-to-Image | COYO [5], EMOVA [8], LAION-2B [63] |
| Stage II: Visual-Language Projector Warmup | 30M | Image-to-Text | LLaVA-150K [37, 36], COYO, EMOVA-Pretrain |
| Stage III: Multimodal Pretraining with Generative and Structural Signals | 35M | Image-to-Text & Editing | UltraEdit [83], AnyEdit [77], SEED-Edit [54] |
| | 3M | Segmentation | RefCOCO-Series [76, 44], RefClef [23], Paco-LVIS [49], PartImageNet [20], DAVIS-2017 [6], Pascal-Part [12], YouTube-VIS2019 [72] |
| | 5M | Dialog / QA | Magpie [69], OpenOrca [32], SCP-116K [42], OpenHermes [58], OPC-SFT-Stage1 [22] |
| Stage IV: Instruction Tuning for Controllable Editing | 7M | Image Editing | EMOVA-SFT, Pixmo [17], M4-Instruct [36], OmniEdit [65], AnyEdit, UltraEdit, InstructPix2Pix [4], MagPie |
| | 2M | Segmentation | ReasonSeg [24], Lisa++ Inst. Seg. & CoT [73], ReVOS [70], Ref-Youtube-VOS [52] |
| | 0.5M | Interactive Editing | COCO-Interactive [82] |

## C  Progressive Dataset Structuring

Each stage in FOCUS adopts dedicated datasets aligned with its training objectives, rather than relying on a uniform corpus across all phases. The dataset distribution and task assignments are summarized in Table 8.

**Stage 1: Visual Quantizer and Diffusion Pretraining.** This stage focuses on learning discrete representations for both semantic abstraction and fine-grained reconstruction. We utilize 45M image-text pairs from COYO, EMOVA, and LAION-2B. COYO contributes OCR-rich, language-aligned samples; EMOVA provides aesthetic-oriented captions; and LAION-2B enhances domain diversity. These datasets collectively support the training of dual visual quantizers and the diffusion decoder.

**Stage 2: Visual-Language Adapter Warmup.** To establish foundational alignment between visual embeddings and the language model, we train on 30M image-text pairs from COYO, EMOVA-Pretrain, and LLaVA-150K. These samples span natural scenes, documents, and instruction-following captions. This stage optimizes only the projection heads and adapter modules, using a language modeling loss while keeping all backbone components frozen.

**Stage 3: Multimodal Pretraining with Generative and Structural Signals.** This stage enhances the model's ability to generate, localize, and reason across modalities. We leverage 35M samples from UltraEdit, SEED-Edit, and AnyEdit for text-guided editing. An additional 3M segmentation

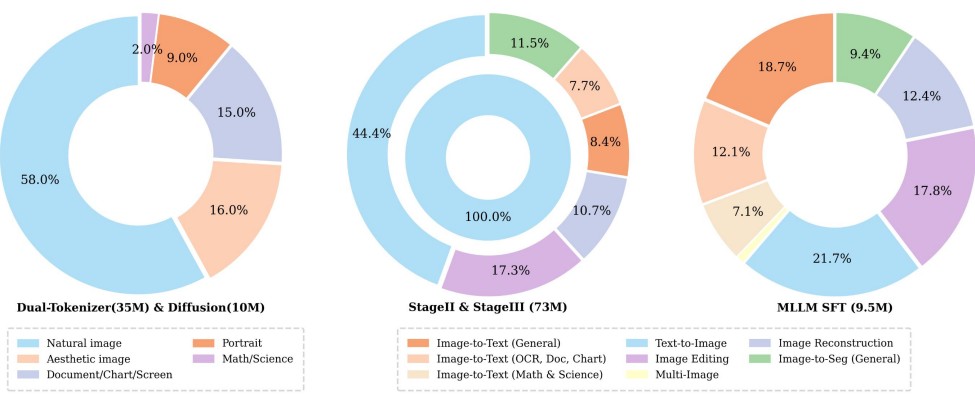

Figure 6: Summary of the data mixture in each stage. Our training data gradually covers a wide range of tasks and various image resoluton.

annotations are sourced from RefCOCO-series, RefClef, and video datasets like DAVIS-2017 and YouTube-VIS2019. For dialogue and question answering, we include 5M samples from Magpie, OpenOrca, SCP-116K, OpenHermes, and OPC-SFT-Stage1. This stage jointly supervises multimodal reasoning, structural grounding, and fine-grained visual tasks.

**Stage 4: Instruction Tuning for Controllable Editing.** The final stage fine-tunes the model for spatially guided, instruction-based editing. We use 7M samples from EMOVA-SFT, Pixmo, M4-Instruct, OmniEdit, AnyEdit, UltraEdit, InstructPix2Pix, and Magpie for complex editing tasks. We incorporate 2M segmentation-centric samples from ReasonSeg, Lisa++, and video segmentation sets like RVOS and RefYoutubeVOS. Additionally, COCO-Interactive provides 0.5M samples for interactive region-level editing. This stage improves instruction following, multi-object control, and editing consistency across visual contexts.

# D    Training Configurations Across Different Stages

Table 9: Training hyperparameters across different stages in FOCUS.

| Settings | Visual Quantizer (Tokenizer) | Diffusion Decoder (Image Reconstruction) | Projector Warmup (Projector Warmup) | Multimodal Pretraining (Seg. Pretrain) | Instruction Tuning (Instruction Tuning) |
|---|---|---|---|---|---|
| Learning Rate | 1e-4 (semantic) 2e-4 (pixel) | 2e-5 | 1e-3 | 2e-5 (Visual encoder, LLM) 1e-3(Mask Decoder) | 2e-5 (Visual encoder, LLM) 2e-6 (Mask Decoder, Diffusion) |
| Batch Size | 256 | 128 | 512 | 128 | 256 |
| Training Steps | 136k (pixel) 28k (semantic) | 220k | 1epoch | 3epoch | 1epoch |
| Image Resolution | 256 to 512 | 512 / 1024 | 256 | 256 / 512 | 512 to 1024 |
| Frozen Modules | Vanilla encoder Hierarchical Encoder | All encoders Codebooks | Visual Encoder, LLM Mask Decoder, Diffusion | Diffusion | Visual Encoder |

We adopt stage-specific training configurations to align with the objectives and resolution requirements of each module, as summarized in Table 9.

**Dual Visual Tokenizer** using a fixed learning rate of 1e-4 and batch size of 256. The training follows three progressive resolution stages: 136k steps at 256×256, 28k steps at 512×512. The semantic branch is optimized with a cosine similarity loss, while the pixel branch is trained with a combination of L1, perceptual, and adversarial losses.

**Diffusion Decoder** using a learning rate of 2e-5 and batch size of 128 for 265k steps. Training employs multi-aspect-ratio cropping up to 1024×1024 resolution. All visual encoders and tokenizers are frozen, and supervision is provided through L2 reconstruction loss.

**LVLM Training** is conducted in three distinct stages.

- **Projector Warmup**: Only the projection heads and gated visual adapters are trained using learning rates of $1e-3$ (projectors), with a batch size of 512 at $256 \times 256$ resolution for 1 epoch.
- **Multimodal Pretraining**: The visual encoder, large language model and mask decoder are optimized jointly. Phase one uses $256 \times 256$ resolution with a batch size of 256 for 1 epoch, and phase two uses $512 \times 512$ with a batch size of 128 for 2 epochs. A learning rate of $2e-5$ is used throughout, and $1e-3$ is used for the mask decoder.
- **Instruction Tuning**: The LLM, segmentation decoder, and diffusion model's attention layers are fine-tuned with learning rates of $2e-5$ (LLM) and $2e-6$ (mask decoder, diffusion), with a batch size of 256 for 1 epoch. Training involves input resolutions randomly sampled from $512 \times 512$ to $1024 \times 1024$ using bucketed cropping.

# E    Metrics.

We evaluate FOCUS across three core tasks using standard benchmarks and metrics. For multimodal understanding, we report accuracy on POPE [29], MM-Vet [78], MMBench [40], SEED [26], and MMMU [79] to assess the model's ability in vision-language reasoning, classification, and grounding. For referring segmentation, we use mean Intersection over Union (mIoU) on RefCOCO, RefCOCO+, RefCOCOg, and gRefCOCO, measuring the overlap between predicted masks and ground truth under language prompts. For controllable image generation and editing, we evaluate fidelity using FID (Fréchet Inception Distance), and alignment using CLIP-based scores (CLIP-I, CLIP-T, CLIP-DIR) to assess visual consistency and instruction compliance. Additional breakdowns from GenAI-bench and GenEval are used for fine-grained control metrics such as object count, color, and spatial placement.

# F    Unified Segmentation Capability of FOCUS

FOCUS is designed as a unified model capable of handling diverse segmentation tasks spanning spatial, temporal, and interactive modalities. We evaluate its generalization and adaptability across five key segmentation scenarios, each highlighting a distinct dimension of its fine-grained visual perception.

## F.1    Contextual Reasoning and Referential Understanding

We evaluate the ability of FOCUS to perform segmentation under semantically complex and referential conditions using the ReasonSeg and ReVOS benchmarks. These tasks require the model to accurately localize and segment objects based on contextual or linguistic descriptions with varying temporal spans. As shown in Table 10, FOCUS achieves leading performance across both benchmarks. On ReasonSeg, it obtains a gIoU of 62.1 and a cIoU of 58.6, surpassing prior methods in semantic segmentation precision. On ReVOS, FOCUS ranks first in all reasoning and referring metrics, including a J&F of 57.2 in reasoning and 58.9 in referring, validating its unified capability in both spatial understanding and temporal reference tracking.

Table 10: Performance on Referring Video Object Segmentation (ReVOS) and ReasonSeg benchmarks. Bold: best; Underlined: second-best.

| Method | Backbone | ReVOS-Reasoning | | | ReVOS-Referring | | | ReVOS-Overall | | | ReasonSeg | |
|---|---|---|---|---|---|---|---|---|---|---|---|---|
| | | J | F | J&F | J | F | J&F | J | F | J&F | gIoU | cIoU |
| LMPM | Swin-T | 13.3 | 24.3 | 18.8 | 29.0 | 39.1 | 34.1 | 21.2 | 31.7 | 26.4 | - | - |
| ReferFormer | Video-Swin-B | 21.3 | 25.6 | 23.4 | 31.2 | 34.3 | 32.7 | 26.2 | 29.9 | 28.1 | - | - |
| LISA-7B | ViT-H | 33.8 | 38.4 | 36.1 | 44.3 | 47.1 | 45.7 | 39.1 | 42.7 | 40.9 | 52.9 | 54.0 |
| LaSagnA-7B | ViT-H | - | - | - | - | - | - | - | - | - | 48.8 | 47.2 |
| SAM4MLLM-7B | Efficient ViT-SAM-XL1 | - | - | - | - | - | - | - | - | - | 46.7 | 48.1 |
| TrackGPT-13B | ViT-H | 38.1 | 42.9 | 40.5 | 48.3 | 50.6 | 49.5 | 43.2 | 46.8 | 45.0 | - | - |
| VISA-7B | ViT-H | 36.7 | 41.7 | 39.2 | 51.1 | 54.7 | 52.9 | 43.9 | 48.2 | 46.1 | 52.7 | **57.8** |
| VISA-13B | ViT-H | 38.3 | 43.5 | 40.9 | 52.3 | 55.8 | 54.1 | 45.3 | 49.7 | 47.5 | - | - |
| HyperSeg-3B | Swin-B | 50.2 | 55.8 | 53.0 | 56.0 | 60.9 | 58.5 | 53.1 | 58.4 | 55.7 | 59.2 | 56.7 |
| FOCUS | ConvNext-L | **51.6** | **56.3** | **57.2** | **56.8** | **61.0** | **58.9** | **54.3** | **59.1** | **56.7** | **62.1** | 58.6 |

### F.2 User-Guided Interactive Perception

To assess FOCUS's adaptability to user-driven segmentation, we evaluate it on the COCO-Interactive benchmark under four input modes: box, point, mask, and scribble. These interactions simulate real-time editing scenarios where users specify partial object regions.

As summarized in Table 11, FOCUS achieves the highest IoU across all modalities, including a box score of 83.4 and a point score of 78.6, demonstrating robust generalization across sparse and dense cues. This consistent performance under varied interaction types highlights FOCUS's potential for responsive and controllable editing applications.

Table 11: Performance on the COCO-Interactive benchmark. IoU (%) under different input modalities.

| Method | Backbone | Box | Scribble | Mask | Point |
|---|---|---|---|---|---|
| SAM | ViT-B | 68.7 | - | - | 33.6 |
| SAM | ViT-L | 71.6 | - | - | 37.7 |
| SEEM | DaViT-B | 42.1 | 44.0 | 65.0 | 57.8 |
| PSALM | Swin-B | 80.9 | 80.0 | 82.4 | 74.0 |
| HyperSeg | Swin-B | 77.3 | 75.2 | 79.5 | 63.4 |
| FOCUS | ConvNext-L | **83.4** | **82.7** | **85.2** | **78.6** |

### F.3 Temporal Consistency in Dynamic Scenes

To evaluate FOCUS's capability in video-level segmentation, we benchmark it on both **Video Object Segmentation (VOS)** and **Video Instance Segmentation (VIS)** tasks. These tasks test the model's ability to consistently track and segment objects across time, even in the presence of occlusions, motion blur, or viewpoint shifts.

Table 12 shows that FOCUS achieves top performance across all datasets, including a J&F score of 79.1 on DAVIS17 and a mAP of 65.7 on YouTube-VIS. Its strong performance on both generic and referring-based video benchmarks confirms its robustness in temporally consistent, instance-aware segmentation.

Table 12: Video segmentation results across DAVIS17 (VOS), Ref-YT, Ref-DAVIS (R-VOS), and YouTube-VIS (VIS). Bold: best; Underlined: second-best.

| Method | Backbone | DAVIS17 (J&F) | Ref-YT (J&F) | Ref-DAVIS (J&F) | YT-VIS (mAP) |
|---|---|---|---|---|---|
| SEEM | DaViT-B | 62.8 | - | - | - |
| OMG-Seg | ConvNeXt-L | 74.3 | - | - | 56.4 |
| ReferFormer | Video-Swin-B | - | 62.9 | 61.1 | - |
| OnlineRefer | Swin-L | - | 63.5 | 64.8 | - |
| UNINEXT | ConvNeXt-L | 77.2 | 66.2 | 66.7 | 64.3 |
| LISA-7B | ViT-H | - | 53.9 | 64.8 | - |
| VISA-13B | ViT-H | - | 63.0 | 70.4 | - |
| VideoLISA-3.8B | ViT-H | - | 63.7 | 68.8 | - |
| HyperSeg-3B | Swin-B | 77.6 | 68.5 | 71.2 | 53.8 |
| FOCUS | ConvNeXt-L | **79.1** | **69.3** | **72.4** | **65.7** |

## G The role of the segmentation mask

Table 13: Ablation study on the effect of segmentation mask guidance on EmuEdit benchmark.

| Method | EmuEdit | | | |
|---|---|---|---|---|
| | DINO($\uparrow$) | CLIP-I($\uparrow$) | CLIP-T($\uparrow$) | CLIP-DIR($\downarrow$) |
| **No Mask Guidance** | 0.812 | 0.866 | 0.268 | 0.113 |
| **Using a General-Purpose Mask** | 0.819 | 0.868 | 0.271 | 0.108 |
| **FOCUS(Ours)** | 0.826 | 0.872 | 0.275 | 0.101 |

To highlight the importance of segmentation masks in our framework, we design three comparative experiments. FOCUS (our full model) utilizes internally generated masks precisely aligned with

language instructions. Baseline 1 removes mask guidance entirely, relying solely on text instructions. Baseline 2 employs masks from an external general-purpose segmentation model (Grounding+SAM2). These experiments demonstrate that precise, instruction-aligned masks are essential for optimal editing performance.

# H   Visualization of Generation Quality

We present a set of images generated by FOCUS based on natural language prompts (see figure 7). These results span various styles including realistic scenes, conceptual designs, and artistic illustrations. They demonstrate the model's ability to produce high-quality and semantically consistent outputs.

Each image is generated solely from text input without using any visual masks or interactive guidance. The visualizations confirm FOCUS's strength in aligning language instructions with fine-grained visual content, showcasing both fidelity and controllability.

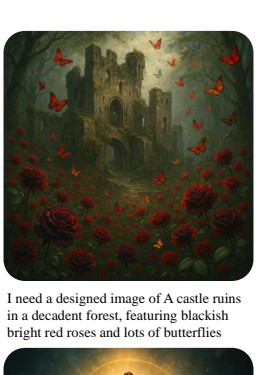 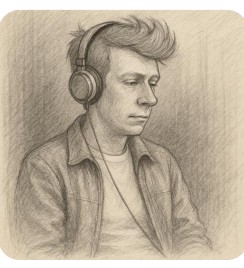 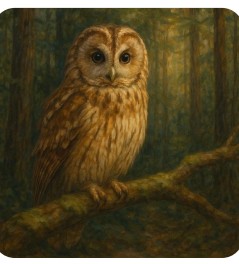 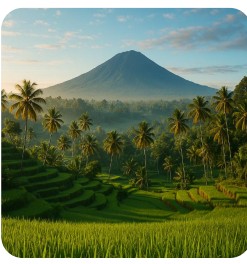

I need a designed image of A castle ruins in a decadent forest, featuring blackish bright red roses and lots of butterflies

Please create a sketched figure of Phillip Fry listening to music in a realistic photo.

I want to see a rendered painting of An owl is perched on the branch in the woods.

Beautiful landscape photography, summer, Indonesia

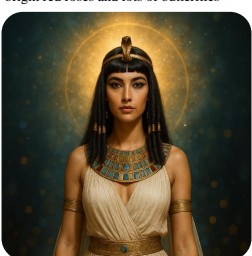 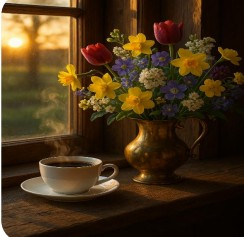 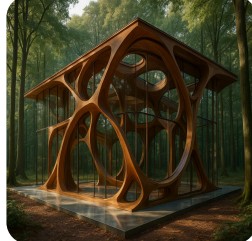 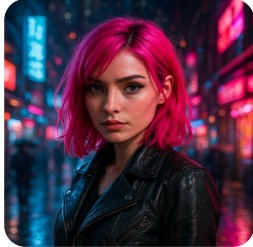

Beautiful surreal symbolism the mesmerizing vision of a Cleopatra Queen of Egypt, mesmerizing brown eyes, black hair and ethereal features, radiating celestial aura, fine art photography, cinematic compositing, authentic, professional by Rorianai style 36k s1000

Real photo of a cup of hot steaming coffee and a brass vase with a large bouquet of spring flowers by an old oak window at sunrise, fine details, rich colors taken with a nikon z6 camera and a shutter speed of 1400 knot. UHD dtm HDR 8k

Architectural parametric pavilion made from wood and glass, with organic cavities, surrounded by a beautiful forest. Dramatic scene, photorealistic, hyperrealistic, raytracing reflections, 8k hd, intricate detail in the style of Frank Lloyde Wright

A detailed high resolution photograph of a captivating cyberpunk girl with vibrant pink hair looking intently at the camera as she stands confidently in a bustling cyberpunk town. The colors include a palette of bold pinks, blues, and purples, with contrasting dark shadows and bright neon highlights.

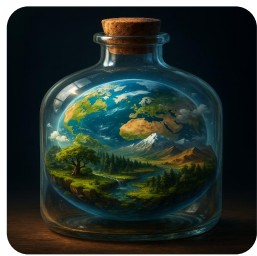 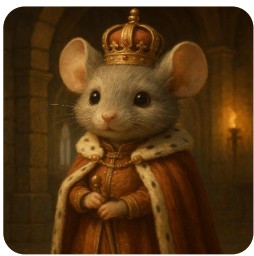 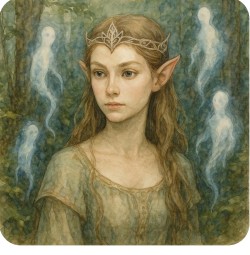 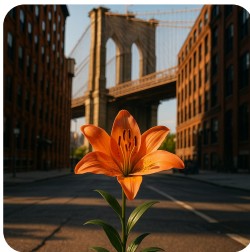

Create a figure of A full biome planet enclosed inside a glass bottle, with intricate details of nature and lighting, presented in ultra high resolution 4k concept art.

Tiny cute adorable mouse dressed as a king in a castle, anthropomorphic, Jean-Baptiste Monge, soft cinematic lighting, 8k, intricate details, portrait, Pixar style character, old fashioned movie style

Can you generate a sketched drawing of A young Elven Princess surrounded by spirit in a forest setting is depicted in an ultra-realistic watercolour portrait with a fantastic touch.

Create a scene of A vibrant and nature-inspired lily flower grows out of the streets of Brooklyn, with the iconic Brooklyn Bridge serving as a breathtaking backdrop.

Figure 7: Visual examples of image generation results from FOCUS given only natural language prompts. The model produces diverse and high-fidelity outputs across a range of styles and scenes, demonstrating strong semantic alignment and visual controllability.

# I  Controllable Image Editing via Segmentation Guidance

To further demonstrate the controllability of FOCUS in localized image editing, we visualize representative examples where the model edits specific regions based on natural language instructions. As shown in Figure 8, the model takes an input image and a user-provided instruction describing the desired modification. It first predicts a segmentation mask that identifies the target region referenced in the text, then uses this mask to guide the generation of an edited output.

This pipeline illustrates the effectiveness of FOCUS in grounding language to spatial regions and applying precise, content-aware modifications. The examples cover diverse scenarios including object transformation, replacement, and contextual adjustment. The results validate that segmentation-aware diffusion guidance enables fine-grained, localized editing while preserving the global scene structure.

Table 14: Prompt schema design for different vision-language tasks. Each row shows how FOCUS handles a specific task by pairing a general instruction (task prompt) with a task-specific condition. This formulation supports unified handling of segmentation, generation, and editing tasks.

| Task Type | Task Instruction Prompt (SI) | Condition Prompt (SC) |
|---|---|---|
| Class-based Segmentation | Please segment all the positive objects by the following candidate categories. | ["person", "dog", "car", "tree", ...] |
| Referring Segmentation | Please segment the target referred to by the language description. | "The man wearing a red hat standing beside the yellow car." |
| Reasoning Segmentation | Please segment the target referred to by the reasoning-based description. | "The object that the man is reaching for in the office." |
| Interactive Segmentation | Please segment according to the given visual reference regions. | Pooled CLIP region features (e.g., clicks, scribbles, boxes) |
| Image Generation | Please generate an image according to the following description. | "A tiny brown dog with white patches, eagerly holding a blue and black Frisbee." |
| Image Editing | Please edit the image according to the following instruction. | "Replace the man in a black jacket with a woman in the same pose." |

# J  Prompt Design for Multi-Task Vision-Language Modeling

To support a wide range of vision-language tasks within a unified framework, FOCUS adopts a structured prompt schema consisting of two components: a task instruction prompt and a condition prompt in Table. 14. The task instruction prompt defines the model objective in natural language, such as segmentation, generation, or editing. The condition prompt provides task-specific contextual information, such as category labels, referential descriptions, or visual cues.

This design enables the model to flexibly adapt to diverse tasks including class-based segmentation, referring and reasoning segmentation, interactive segmentation with visual cues, text-to-image generation, and fine-grained image editing. By standardizing task formulation through prompt schema, FOCUS achieves better generalization across modalities and applications.

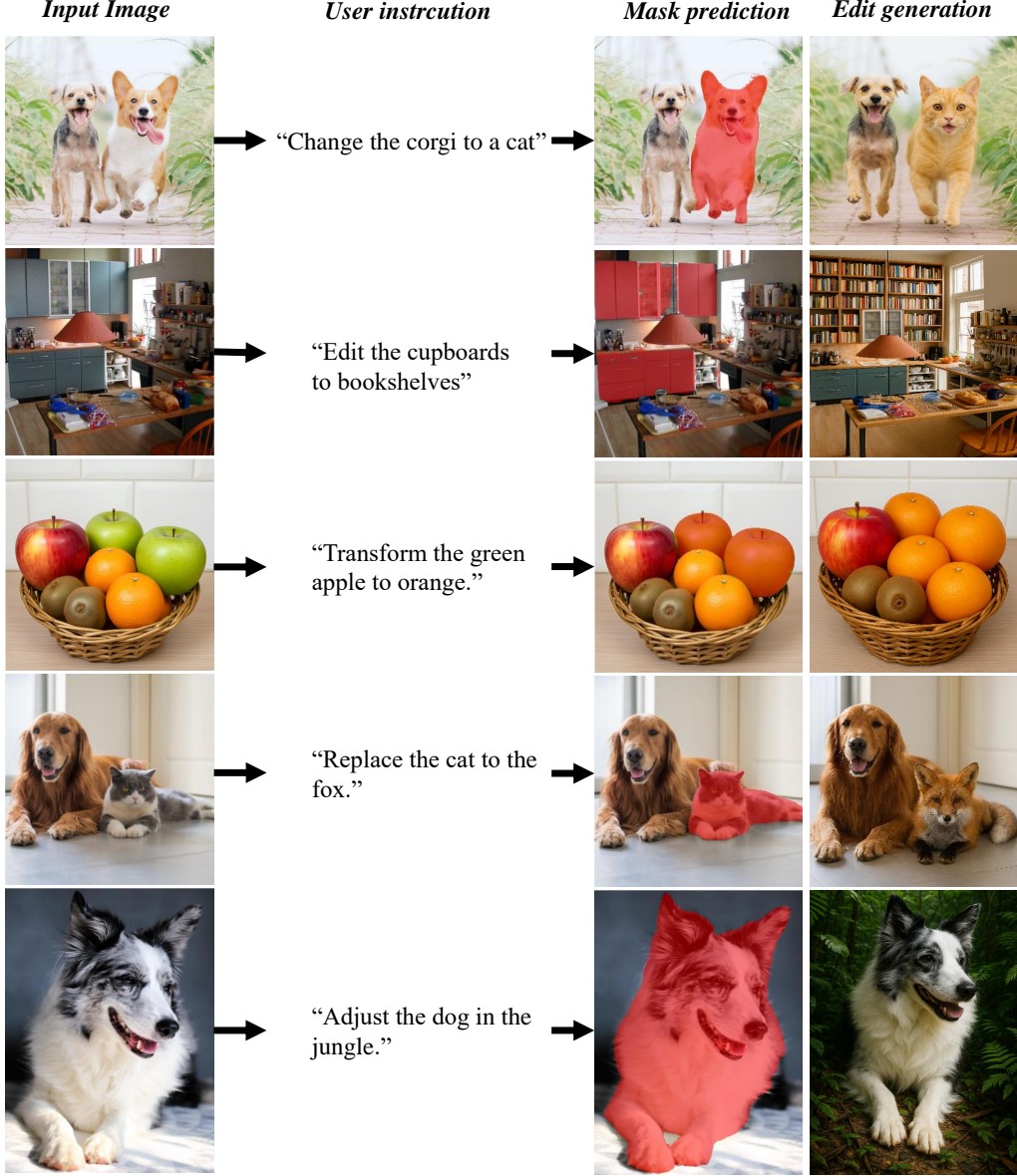

Figure 8: Visualization of controllable image editing results. Given an input image and a user instruction, FOCUS first predicts a spatial mask corresponding to the referential target, then performs localized generation to edit the specified region. The examples demonstrate accurate region identification and high-fidelity edits aligned with the instruction.

