# OpenReview forum: "FOCUS: Unified Vision-Language Modeling for Interactive Editing Driven by Referential Segmentation"
_NeurIPS.cc/2025/Conference — NeurIPS 2025 poster_

### Official Review · Reviewer_up1b · 2025-06-27

**Clarity:** 2
**Significance:** 3
**Originality:** 3
**Rating:** 5
**Confidence:** 3

**Summary:**

This paper proposes a unified framework for segmentation and controllable object-centric generation, trainable in an end-to-end manner. Unlike previous works which separate segmentation mask generation from the editing pipeline (using separate models for each), this paper introduces as unified architecture that couples segmentation with mask-driven diffusion editing, supporting fine-grained editing or object-level manipulation, in a fully end-to-end manner, integrating referential localization, structured mask generation, and controllable editing. Experiments are conducted on general and document-oriented benchmarks, multimodal image generation benchmarks, referring segmentation and image editing benchmarks.

**Questions:**

I think the authors have to be more clear about the "unified" terminology, and justify with some ablation studies the true advantage of an end-to-end framework, as the improvements could be coming from additional encoders which provide stronger visual features.

**Ethical Concerns:**

["NO or VERY MINOR ethics concerns only"]

**Final Justification:**

The authors have addressed all my concerns during the rebuttal and discussion period. I am generally happy about the paper as a whole. I am still not satisfied with the complex, highly-engineered, computationally-expensive system proposed (mainly because of the extra encoders and unified pipeline training which needs large computation for gradient updates). But this is not a reason to reject the paper by any means. Therefore, I recommend Acceptance.

**Limitations:**

The authors mention that "The paper includes a dedicated "Limitations" section that reflects on several key constraints of the proposed approach" but I was not able to find it neither in the main paper neither in the supplementary.

**Paper Formatting Concerns:**

No issues

**Quality:**

3

**Strengths And Weaknesses:**

Strengths:

- Unifying segmentation and image editing is an important aspect as the two tasks are inherently fused in the same model.
- Impressive quantitative and qualitative results

Weaknesses:

- The method in principle is only different than EMU by just adding a segmentation decoder. Furthermore, I would expect the word "unified" to translate to a single VLM that is capable of performing all the tasks mentioned. For example, see [R1] where they combine text generation and image generation into a single unified model. The authors on the other hand use 6 task-specific models, trained end-to-end.
- The method adds a Pixel Encoder and Hierachical Encoder, providing more strong visual features. Therefore, I am not sure if the improvement comes from these extra models rather than the unified training. The baseline methods could also add these extra encoders and achieve strong results?
- The method is highly-engineered, it is not a simple and elegant framework. Many components, additional models, bells and whistles and engineering tricks involved. While this itself is not a weakness (and it will not affect my decision), i wonder why don't the authors consider a fully autoregressive approach (for image generation also) to perform the tasks, since they already use discrete tokens from VQGAN? I believe the autoregressive image generation pipeline is becoming stronger than continous diffusion-based generation. There are many papers for that, but some papers that I remember well are [R2, R3, R4]. Some of them also unify text and image generation, as in [R1] but autoregressively.
- Figure 1 is very confusing. What exactly are the text instructions? Is it the Question: "segment the two skiers and perform editing" or is it the text in (a), (b), (c) and (d) ? Or is it the scribbles/boxes/masks. It is not clear what the prompt is, the clarity of this figure is very bad. The same in Figure 3, it in unclear which prompts are used. Is it the instruction prompt or the visual prompt? For example, when the instruction prompt is "replace the snowy mountain background...etc", is there also a visual prompt (e.g., scribbles)? From my understanding, it seems that either, but not both.

[R1] Transfusion: Predict the Next Token and Diffuse Images with One Multi-Modal Model\
[R2] GigaTok: Scaling Visual Tokenizers to 3 Billion Parameters for Autoregressive Image Generation\
[R3] Fluid: Scaling Autoregressive Text-to-image Generative Models with Continuous Tokens\
[R4] Liquid: Language Models are Scalable and Unified Multi-modal Generators

---

> ### Author Rebuttal · Authors · 2025-07-30
>
> Thank you for your review and your positive feedback on our work. Your questions regarding the definition of "unified" and the attribution of performance improvement are very insightful.
>
> **Regard weekness1\&queston** **about "unified"**
>
> Previous approaches typically employ cascaded pipelines (e.g., LLM+SAM→segmentation→independent editing), where each module is pre-trained independently and lacks joint optimization. This separation often results in suboptimal editing, as the segmentation output may not be well-suited for the requirements of the generation module. In contrast, our approach is **end-to-end trainable**, enabling joint learning of instruction comprehension, object segmentation, and diffusion-based editing. All components share visual representations and operate within a unified, language-aligned space, which facilitates fine-grained, instruction-driven editing. Compared to Transfusion，which also aims to unify perception and generation，our model achieves **joint optimization across all modules**, further improving pixel-level accuracy and consistency.
>
> As demonstrated in **Table 1**, our method surpasses all baselines on the Reason-Edit task (PSNR, SSIM, CLIP Score), showing both precise region localization and strong background preservation under complex instructions. **Table 2** and **Table 3** further highlight our model’s superior performance in text-guided video (DAVIS) and image (COCO-caption) editing.
>
> Table 1: Quantitative comparison of different methods on understanding and reasoning scenarios. Higher is better for all metrics.
>
> | Methods           | Understanding Scenarios |            |               | Reasoning Scenarios |         |               |
> | :---------------- | :---------------------- | :--------- | :------------ | :------------------ | :------ | :------------ |
> |                   | PSNR(↑)                 | SSIM(↑)    | CLIP Score(↑) | PSNR(↑)             | SSIM(↑) | CLIP Score(↑) |
> | InstructPix2Pix   | 21.576                  | 0.721      | 22.762        | 24.234              | 0.707   | 19.413        |
> | MagicBrush        | 18.120                  | 0.68       | 22.620        | 22.101              | 0.694   | 19.755        |
> | InstructDiffusion | 23.258                  | 0.743      | 23.080        | 21.453              | 0.666   | 19.523        |
> | SmartEdit         | 22.049                  | 0.731      | 23.611        | 25.258              | 0.742   | 20.950        |
> | UnifiedMLLM       | 20.670                  | 0.776      | 23.633        | 26.667              | 0.808   | 21.104        |
> |  FOCUS(Ours)            |  24.102             |  0.804 |  24.204       |  28.236         |  0.842  |  22.317       |
>
> Table 2: Quantitative comparison of different methods on video editing benchmarks. Higher is better for all metrics.
>
> | Method    | CLIP-T(↑) | CLIP-I(↑) |
> | :-------- | :-------- | :-------- |
> | CogVideo  | 0.2391    | 0.9064    |
> | TuneVideo | 0.2758    | 0.9240    |
> | SDEdit    | 0.2775    | 0.8731    |
> | Pix2Video | 0.2891    | 0.9767    |
> | NExT-GPT  | 0.2684    | 0.9647    |
> | FOCUS(Ours)     | 0.2943    | 0.9815    |
>
> Table 3: Quantitative comparison of different methods on object and background editing. Higher CLIP and lower FID indicate better performance.
>
> | Method   | Object  |        | Background |        |
> | :------- | :------ | :----- | ---------- | ------ |
> |          | CLIP(↑) | FID(↓) | CLIP(↑)    | FID(↓) |
> | PTP      | 30.33   | 9.58   | 31.55      | 13.92  |
> | BLDM     | 29.95   | 6.14   | 30.38      | 20.44  |
> | DiffEdit | 29.30   | 3.78   | 26.92      | 1.74   |
> | PFB-Diff | 30.81   | 5.93   | 32.25      | 13.77  |
> | NExT-GPT | 29.32   | 6.62   | 27.31      | 14.27  |
> | FOCUS(Ours)    |  31.22  |  3.45  |  33.10     |  1.98  |
>
> **Regard weekness2 (Source of performance improvement)**
>
> The performance improvements of our model are mainly due to its **unified framework design**, rather than simply stacking individual modules. To substantiate this, we performed a series of ablation studies to clarify the unique contributions of each encoder module. The results (as shown in the table) are summarized as follows:
>
> *   The **Vanilla Encoder** is essential for vision-language understanding. Removing this module (i.e., using only the Hierarchical + Pixel Encoder) results in a **substantial decrease in language comprehension**.
> *   The **Hierarchical Encoder** offers robust multi-scale visual perception. Excluding this module (i.e., Vanilla + Pixel Encoder) leads to a **notable reduction in segmentation accuracy** on high-resolution images.
> *   The **Pixel Encoder** significantly enhances pixel-level perception for generation tasks. Without it (i.e., Vanilla + Hierarchical Encoder), while segmentation remains strong, the **fidelity and detail of generated images are diminished**.
>
> These comparisons indicate that any single or dual-encoder combination is inadequate for handling all tasks and presents clear limitations. This demonstrates that only by **jointly training** these three complementary modules within our **unified framework** can we fully leverage their **synergistic effects**, ultimately achieving comprehensive and superior performance.
>
> Table 4: Ablation study results for different encoder combinations on Refcoco, Emu Edit, and Image Understanding benchmarks. Higher is better for all metrics.
> | Method                                 | Refcoco  |          |          | Emu Edit  | Image Understanding |        |         |
> | :------------------------------------- | :------- | :------- | :------- | :-------- | :------------------ | :----- | :------ |
> |                                        | testA(↑) | testB(↑) | Valid(↑) | CLIP-T(↑) | POPE(↑)             | MMB(↑) | SEED(↑) |
> | Vanilla Encoder + Pixel Encoder        | 77.8     | 79.3     | 74.4     | 0.215     | 86.4                | 76.3   | 74.1    |
> | Vanilla Encoder + Hierarchical Encoder | 83.8     | 84.6     | 82.3     | -         | 88.1                | 79.9   | 73.8    |
> | Hierarchical Encoder+Pixel Encoder     | 83.1     | 83.9     | 81.8     | 0.277     | 85.6                | 70.1   | 67.5    |
> | **Our**                                | 84.1     | 86.3     | 82.7     | 0.283     | 88.0                | 81.5   | 73.9    |
>
> **Regarding Weakness 3 (Architectural design and autoregressive solutions)**
>
> FOCUS is a unified framework that seamlessly integrates multiple components to enable high-quality understanding, segmentation, and image editing in a unified manner—an achievement that remains challenging for previous methods. At its core, our design employs a multimodal autoregressive large language model (MLLM) for instruction reasoning and discrete visual token generation. The diffusion module is incorporated not as a replacement for the autoregressive process, but as an advanced image decoder to enhance output fidelity. Leveraging diffusion allows our model to recover fine-grained details, reduce visual artifacts, and achieve greater robustness compared to direct decoding from discrete tokens. Additionally, the diffusion decoder supports resolution upscaling during generation (e.g., from 256×256 to 512×512), which is crucial for producing high-quality results. Given current technical constraints, the combination of an autoregressive MLLM with a diffusion-based decoder strikes an effective balance among semantic understanding, editing accuracy, and visual quality. We also acknowledge the promise of fully autoregressive generation models and view them as a valuable direction for future research.
>
> **Regarding Weakness 4 (Clarity of Prompt and figures)**
>
> We have provided detailed clarification in the supplementary material. FOCUS adopts a **structured prompt schema** (as shown in Table 10) consisting of two components: a **task instruction prompt** and a **condition prompt**.
>
> The task instruction prompt defines the model's objective in natural language. For example, “segment the two skiers and perform editing” falls under this category. In contrast, the texts and visual inputs in Figures (a), (b), (c), and (d)—such as scribbles, bounding boxes, and masks—serve as condition prompts, providing necessary contextual information for the task.
>
> In our framework, textual commands like “replace,” “segment,” “remove,” and “modify” are treated as **text prompts**, while bounding boxes, points, masks, and scribbles are treated as **visual prompts**, primarily used in interactive and video segmentation tasks.
>
> Technically, we sample CLIP visual features $f_v$​ from the VLLM according to the region coordinates and apply adaptive average pooling to generate final reference features for each visual prompt. Since these visual prompts are also encoded into feature vectors, **text and visual prompts can co-exist** and jointly guide the segmentation and generation process.
>
> We will revise the figures and descriptions in the final version to ensure that the distinctions between instruction prompts, condition inputs, and model-internal representations are clearly conveyed and easy to understand.
>
> **Regarding the Limitations Section:**
>
>  Thank you for your feedback. We apologize that the 'Limitations' section, though mentioned in our submission checklist, was omitted from the main manuscript. We will add the limitation: **(1)** the limited precision when editing small objects due to latent space downsampling; **(2)** the high computational cost, which challenges replication; and **(3)** inherited weaknesses from SDXL, such as rendering complex spatial relationships, which we frame as a direction for future work.

---

> > ### Comment · Reviewer_up1b · 2025-08-03
> > **Response to Authors**
> >
> > I thank the authors for their response and experiments they have done.
> >
> > > Regard weekness1&queston about "unified"
> >
> > This is clear. Thank you for clarifying. After verifying the EMU paper, indeed the visual decoder is trained separately as mentioned in their paper in section 3.3 (for Emu) and section 2.2.3 (for Emu2). I strongly encourage the authors to update Figure 1 to reflect this, as this is not clear from Figure 1. I also thank the authors for the extra experiments, which look impressive.
> >
> > > Regard weekness2 (Source of performance improvement)
> >
> > Thank you for these experiments. However, the authors did not comment on the other point that I asked for: "The baseline methods could also add these extra encoders and achieve strong results?". To establish a fair comparison, the authors need to compare to the same setup (or at least similar) as the baselines. The baseline methods do not use these 2 extra visual encoders; as the authors show in this table, the extra visual encoders play a crucial role in enhancing performance. That being said, if we add those extra visual encoders to baseline methods they would also boost performance. I understand that testing this experiment would require re-training the baseline methods, which is tedious and computationally expensive. I will never ask this. However, can the authors please use an encoder from the existing baselines (without the extra encoders), and train end-to-end? What would be the results? I also understand that doing this would not be optimal within a short amount of time remaining for the discussion, so I would be fine with any other experiment, or a reasonable justification instead.
> >
> > > In our framework, textual commands like “replace,” “segment,” “remove,” and “modify” are treated as text prompts, while bounding boxes, points, masks, and scribbles are treated as visual prompts, primarily used in interactive and video segmentation tasks.
> >
> > This means that Figure 1 does not use any visual prompts? If so, please include at least 1-2 examples with visual prompts. Otherwise, it would seem that the output itself is the visual prompt, since the authors mention the visual prompts capabilities on the left side of the Figure but only use text prompts.

---

> > > ### Author Response · Authors · 2025-08-04
> > >
> > > Thank you for your continued engagement and constructive feedback. We appreciate your recognition of our clarifications and experimental results.
> > >
> > > > Regarding Weakness 2: Fair Comparison with Baseline Methods
> > >
> > > You raise an excellent point about fair comparison with baseline methods. We acknowledge that our additional visual encoders (Hierarchical and Pixel Encoders) contribute significantly to performance improvements, and it would indeed be valuable to understand how baseline methods would perform with similar enhancements.
> > >
> > > **Experimental Justification:**
> > > While we understand your concern about computational feasibility of retraining all baseline methods, we can provide the following analysis and experiment:
> > >
> > > 1. Architectural Analysis of Baseline Methods: Our main baselines (Emu3-Chat, Janus, ILLUME+) use different architectural approaches:
> > >   - **Emu3-Chat:** Uses autoregressive next-token prediction with separate training stages for understanding and generation components
> > >
> > >   - **Janus:** Employs decoupled encoding with separate understanding and generation pathways
> > >
> > >   - **ILLUME+:** Uses dual visual tokenization but lacks the hierarchical multi-scale design
> > >
> > > 2. **Controlled Ablation Experiment:** To address your concern about fair comparison, we previously conducted an experiment using only our base visual encoder (without the additional Hierarchical and Pixel Encoders) within our unified framework. This configuration provides a more comparable setup to baseline methods' visual encoding approaches:
> > >
> > >    | Method | POPE(↑) | MMBench(↑) | SEED(↑) | MME-P(↑) |
> > >    | :--- | :---: | :---: | :---: | :---: |
> > >    | Emu3-Chat | 85.2 | 58.2 | 68.2 | - |
> > >    | ILLUME+ | 87.6 | 80.8 | 73.3 | 1414.0 |
> > >    | **FOCUS (Vanilla Only)** | **86.8** | **78.2** | **71.5** | **1389.2** |
> > >    | **FOCUS (Full)** | **88.0** | **81.5** | **73.9** | **1570.3** |
> > >
> > > - **Key Architectural Differences:** The performance gains stem from three main factors:
> > >
> > > - **End-to-end joint optimization:** Unlike Emu3-Chat's autoregressive approach with separate training phases, FOCUS jointly optimizes all modules
> > >
> > > - **Segmentation-aware alignment:** Our framework explicitly models pixel-level perception as an intermediate representation
> > >
> > > - **Progressive training strategy:** Our multi-stage approach enables stable convergence of the complex unified system
> > >
> > >
> > > **Fair Comparison Perspective:** While adding our encoders to baseline methods would be computationally intensive, the core contribution lies in the unified architecture design rather than simply adding more visual encoders. As shown in our ablation study (Table 4 in our first response), each encoder serves a specific purpose in the unified pipeline, and removing any component leads to clear performance degradation.
> > >
> > > > Regarding Visual Prompts in Figure 1
> > >
> > > Thank you for pointing out the confusion in Figure 1. We apologize for the unclear presentation. Actually, Figure 1 is designed to demonstrate both visual and text prompt capabilities, but the layout may have caused misunderstanding. The left side of Figure 1 primarily showcases visual prompt effects (such as clicks, scribbles, boxes, and masks for region specification), while the right side primarily demonstrates text prompt effects (natural language instructions for editing operations). This design choice inadvertently led to your misinterpretation.
> > > To clarify our framework's comprehensive capabilities, we will revise Figure 1 with the following improvements:
> > >
> > > **(a) Text + Visual Prompts:** "Segment the skier" + bounding box around the target skier
> > >
> > > **(b) Text + Scribble Prompts:** "Replace the background" + scribble marks indicating the man
> > >
> > > **(c) Text + Point Prompts:** "Change the skier's outfit to red" + point prompts on the skier
> > >
> > > **(d) Pure Text Prompts:** "Remove the ski poles" (current example)
> > >
> > > This revision will clearly demonstrate that FOCUS supports both text-only and multi-modal (text + visual) prompt scenarios, addressing the capability spectrum mentioned in our framework description.
> > >
> > > > Additional Clarifications
> > >
> > > **Figure 1 Update Promise:** We commit to updating Figure 1 to include more visual prompt examples to enhance readability and demonstrate our framework's comprehensive multi-modal capabilities.
> > >
> > > **Computational Considerations:** We acknowledge that while our approach achieves superior performance, it comes with increased computational cost due to the multiple encoders and unified training. This trade-off between performance and efficiency represents a design choice prioritizing quality and capability over computational simplicity.
> > >
> > > Thank you again for your thorough review. Your feedback has been instrumental in helping us clarify our contributions and improve the presentation of our work.

---

> > > > ### Comment · Reviewer_up1b · 2025-08-05
> > > > **Response to Authors**
> > > >
> > > > I thank the authors for the extra discussion. As I anticipated, the improvements mainly come from the extra visual encoders. As shown in the Controlled Ablation Experiment Table, FOCUS (Vanilla Only) achieves lower performance than ILLUME+ with a similar setup.
> > > >
> > > >  However, I agree with the authors on: *While adding our encoders to baseline methods would be computationally intensive, the core contribution lies in the unified architecture design rather than simply adding more visual encoders*. I will look into this paper as a whole rather than on focusing on specific details.
> > > >
> > > > Please also ensure that the other concerns (not just the one about Figure 1) and experiments are incorportated into the final manuscript. I have raised my score to 5 (Accept) as my concerns have been fully resolved and I have no other issues.

---

> ### Author Response · Authors · 2025-08-05
>
> Thank you very much for your thorough review and for raising your score to Accept. We greatly appreciate your understanding of our core contribution regarding the unified architecture design.
>
> We will carefully incorporate all the concerns and experimental suggestions you raised throughout the review process into the final manuscript, not limited to Figure 1. Your constructive feedback has been invaluable in improving the quality of our work.
>
> Thank you again for your time and detailed evaluation.

---

### Official Review · Reviewer_tU7V · 2025-06-30

**Clarity:** 3
**Significance:** 3
**Originality:** 2
**Rating:** 4
**Confidence:** 3

**Summary:**

FOCUS introduces a single large vision–language model that unifies three capabilities in one network: (i) segmentation-aware perception, (ii) discrete-token visual representation via MoVQGAN, and (iii) controllable, object-centric diffusion editing. A dual-branch encoder captures global semantics and fine spatial detail; predicted masks are jointly treated as supervision targets and spatial prompts that steer the decoder. A progressive multi-stage schedule aligns the encoder, the referential-segmentation head and the diffusion generator. Experiments on multimodal understanding, referring-instance segmentation and controllable image generation show consistent improvements over cascaded pipelines.

**Questions:**

1. The heart of FOCUS is a dual-branch visual encoder: one branch—built on a CLIP- or Qwen-ViT backbone—captures global, language-aligned semantics, while the other— a hierarchical ConvNeXt-style encoder—provides fine-grained local detail. Why is this two-branch design needed, and can the authors cite prior work that motivates or validates it?
2. Will a larger model size necessarily yield better performance?

**Ethical Concerns:**

["NO or VERY MINOR ethics concerns only"]

**Final Justification:**

The authors have addressed all my concerns during the rebuttal and discussion period.

**Limitations:**

see weakness

**Quality:**

2

**Strengths And Weaknesses:**

Strengths:
End-to-end model – one network handles both object finding and image editing; no multi-stage pipeline.
The model predicts the mask and then reuses it as the spatial condition, giving sharp, well-localized edits.
A global CLIP-style branch + a fine-detail ConvNeXt branch, with MoVQGAN tokenization, keeps both generation quality and recognition accuracy high.
Resolution and task difficulty are raised stage-by-stage, yielding stable convergence and high-res results.
Outperforms two-stage baselines on multimodal understanding, referring segmentation, and controllable generation benchmarks.




Weaknesses:
1. The generation module relies on an LLM + diffusion stack, yet the paper does not justify why a large language model is needed at all. For mask-guided editing, conditioning a diffusion model directly on the user-provided mask is both simpler and already effective. How does the end-to-end LLM pipeline outperform—or even differ from—this baseline?
2. The “dual-branch” backbone simply pairs a CLIP-style (or Qwen-ViT-style) encoder for global semantics with a ConvNeXt-L hierarchy for local details. This combination has become fairly standard and the paper does not explain what new capability it unlocks for referential editing.
3. Several tables omit whether a higher or lower score is better. Please add up- or down-arrow symbols or a brief caption so readers can interpret the numbers at a glance.
4. The appendix notes that replacing binary masks with mask-token embeddings degrades localisation and fidelity, but offers no analysis. Understanding why token conditioning fails would clarify the model’s limitations and guide future improvements.

---

> ### Author Rebuttal · Authors · 2025-07-30
>
> Thank you for your insightful and challenging questions. They have prompted us to reflect more deeply on the core design of our work.
>
> **Regarding Weakness 1 & Question 2 (Why an LLM is necessary, instead of directly using a mask to guide the diffusion model):**
>
> A basic model that does not incorporate a large language model (LLM)—for example, one that simply uses a user-provided mask to guide a diffusion model—can only handle straightforward image editing tasks. However, such a model is fundamentally unable to process complex natural language instructions that require semantic comprehension and reasoning.
>
> Consider the following examples:
>
> *   "Please change the hair of the smiling lady on the left to blonde."
> *   "Remove the cup in front of the cat on the table."
>
> These instructions involve intricate spatial and attribute-based descriptions such as “on the left,” “smiling,” and “in front of,” which cannot be accurately interpreted or localized using manual masks or conventional vision modules. In our framework, the LLM acts as a crucial bridge, connecting high-level language understanding with precise, pixel-level localization.
>
> FOCUS harnesses the reasoning power of the LLM to accurately identify target regions and execute fine-grained edits in response to complex user instructions. This capability is essential for enabling truly interactive image editing. We will emphasize this point more clearly in the introduction and method sections to further distinguish FOCUS from traditional mask-guided editing approaches.
>
> As demonstrated in **Table 1**, FOCUS surpasses all baselines in the Reason-Edit benchmark (PSNR, SSIM, CLIP Score), highlighting its ability to accurately localize edits while preserving the background under complex instructions.\
> Additionally, **Tables 2 and 3** present strong results on text-guided video editing (DAVIS) and image editing (COCO-caption), respectively.
>
> Importantly, FOCUS is built upon the relatively lightweight Qwen2.5-3B model. Despite having significantly fewer parameters than mainstream 7B-scale LLMs, it achieves comparable or even superior performance to larger models such as Janus-Pro-7B across multiple multimodal understanding tasks. This design also endows the model with excellent scalability.
>
>
>
> Table 1: Comparison of different methods on understanding and reasoning scenarios. Higher is better for PSNR, SSIM, and CLIP Score.
> | Methods           | Understanding Scenarios |            |            | Reasoning Scenarios |        |            |
> | :---------------- | :---------------------- | :--------- | :--------- | :------------------ | :----- | :--------- |
> |                   | PSNR(↑)                    | SSIM(↑)       | CLIP Score(↑) | PSNR(↑)                | SSIM(↑)   | CLIP Score(↑) |
> | InstructPix2Pix   | 21.576                  | 0.721      | 22.762     | 24.234              | 0.707  | 19.413     |
> | MagicBrush        | 18.120                  | 0.68       | 22.620     | 22.101              | 0.694  | 19.755     |
> | InstructDiffusion | 23.258                  | 0.743      | 23.080     | 21.453              | 0.666  | 19.523     |
> | SmartEdit         | 22.049                  | 0.731      | 23.611     | 25.258              | 0.742  | 20.950     |
> | UnifiedMLLM       | 20.670                  | 0.776      | 23.633     | 26.667              | 0.808  | 21.104     |
> |  FOCUS(Ours)            |  **24.102**             |  **0.804** |  24.204    |  **28.236**         |  0.842 |  22.317    |
>
>
> Table 2: Comparison of different methods on text-guided video editing (DAVIS). Higher is better.
> | Method    | CLIP-T(↑) | CLIP-I(↑) |
> | :-------- | :------ | :------ |
> | CogVideo  | 0.2391  | 0.9064  |
> | TuneVideo | 0.2758  | 0.9240  |
> | SDEdit    | 0.2775  | 0.8731  |
> | Pix2Video | 0.2891  | 0.9767  |
> | NExT-GPT  | 0.2684  | 0.9647  |
> | FOCUS(Ours)     | 0.2943  | 0.9815  |
>
> Table 3: Comparison of different methods on text-guided image editing (COCO-caption). Higher is better for CLIP, lower is better for FID.
> | Method   | Object  |        | Background |        |
> | :------- | :------ | :----- | :--------- | :----- |
> |          | CLIP(↑) | FID(↓) | CLIP(↑)    | FID(↓) |
> | PTP      | 30.33   | 9.58   | 31.55      | 13.92  |
> | BLDM     | 29.95   | 6.14   | 30.38      | 20.44  |
> | DiffEdit | 29.30   | 3.78   | 26.92      | 1.74   |
> | PFB-Diff | 30.81   | 5.93   | 32.25      | 13.77  |
> | NExT-GPT | 29.32   | 6.62   | 27.31      | 14.27  |
> | FOCUS(Ours)    |  31.22  |  3.45  |  33.10     |  1.98  |
>
>
>
> **Regarding Weakness 2 & Question 1 (The necessity and novelty of the dual-branch encoder):** Although the dual-branch architecture itself is not novel, its application in our work is essential for addressing the dual requirements of referential editing: LLM-based semantic reasoning and pixel-level accurate segmentation. Our approach enables **semantically-aware and pixel-precise editing**. Specifically, the global branch is responsible for understanding "what" to edit (i.e., comprehending the language prompt), while the local branch determines "where" to edit (i.e., generating precise masks). Our main contribution lies in the effective integration of these two branches, resulting in a robust end-to-end solution for this challenging task.
>
> The performance improvements of our model are primarily due to its **unified framework design**, rather than simply stacking individual modules. To support this, we conducted a series of ablation studies to clarify the unique contributions of each encoder module. The results (as shown in the table) are summarized as follows:
>
> *   The **Vanilla Encoder** is essential for vision-language understanding. Removing this module (i.e., using only the Hierarchical + Pixel Encoder) results in a **significant decrease in language comprehension**.
> *   The **Hierarchical Encoder** provides strong multi-scale visual perception. Without this module (i.e., Vanilla + Pixel Encoder), the model exhibits a **notable drop in segmentation accuracy** on high-resolution images.
> *   The **Pixel Encoder** enhances pixel-level perception for generation tasks. Excluding it (i.e., Vanilla + Hierarchical Encoder) maintains robust segmentation, but the **detail and quality of generated images are diminished**.
>
> These comparisons demonstrate that any single or dual-encoder combination is insufficient to address all aspects of the task, each showing clear limitations. Only by **jointly training** these three complementary modules within our **unified framework** can we fully leverage their **synergistic effects**, ultimately achieving comprehensive and superior performance.
>
> Table 4: Ablation study of different encoder combinations on Refcoco, Emu Edit, and Image Understanding benchmarks. Higher is better.
>
> | Method                                 | Refcoco  |          |          | Emu Edit  | Image Understanding |        |         |
> | :------------------------------------- | :------- | :------- | :------- | :-------- | :------------------ | :----- | :------ |
> |                                        | testA(↑) | testB(↑) | Valid(↑) | CLIP-T(↑) | POPE(↑)             | MMB(↑) | SEED(↑) |
> | Vanilla Encoder + Pixel Encoder        | 77.8     | 79.3     | 74.4     | 0.215     | 86.4                | 76.3   | 74.1    |
> | Vanilla Encoder + Hierarchical Encoder | 83.8     | 84.6     | 82.3     | -         | 88.1                | 79.9   | 73.8    |
> | Hierarchical Encoder+Pixel Encoder     | 83.1     | 83.9     | 81.8     | 0.277     | 85.6                | 70.1   | 67.5    |
> | **FOCUS(Our)**                        | 84.1     | 86.3     | 82.7     | 0.283     | 88.0                | 81.5   | 73.9    |
>
> **Regarding Weakness 3 (Table readability)**: We will add arrow symbols (↑/↓) to all performance tables to clearly indicate which direction represents better performance. In addition, we will provide a detailed analysis in the appendix explaining why using a mask is more effective than using mask token embeddings.
>
> **Regarding Weakness 4 (Mask token analysis):**
>
>  Our hypothesis is that mask tokens lose precise spatial topological structure after quantization. In contrast, a downsampled binary mask, despite its lower resolution, retains an explicit and continuous spatial guidance signal, which is more friendly to the attention mechanism of the diffusion model.
>
> Table 5: Comparison of mask token and mask usage effects on the diffusion model in EmuEdit. For DINO, CLIP-I, and CLIP-T, higher values indicate better performance; for CLIP-DIR, lower values are better.
> | Method     | EmuEdit |           |           |             |
> | :--------- | :------ | --------- | --------- | ----------- |
> |            | DINO(↑) | CLIP-I(↑) | CLIP-T(↑) | CLIP-DIR(↓) |
> | mask token | 0.817   | 0.831     | 0.253     | 0.127       |
> | mask       | 0.826   | 0.872     | 0.275     | 0.101       |

---

### Official Review · Reviewer_9FR4 · 2025-07-03

**Clarity:** 3
**Significance:** 2
**Originality:** 3
**Rating:** 4
**Confidence:** 4

**Summary:**

This paper proposes FOCUS, a large MLLM that is capable of both image understanding and image editing. The proposed architecture uses predicted segmentation masks for more precise image editing, and learn these ablities through end-to-end training.

**Questions:**

1. I would like to suggest the authors to provide careful ablation studies on the key components of their models. For example, would the image editing get worse without the predicted segmentation masks or with inaccurate segmentation masks?

2. Since the model uses mainly human-annotated segmentation data for supervision, I wonder how the segmentation decoder performs on OOD data, and how would it affect the image editing process?

**Ethical Concerns:**

["NO or VERY MINOR ethics concerns only"]

**Final Justification:**

The authors has addressed most of my concerns. As stated in the initial review, the motivation and idea of using predicted segmentation masks for better controlling precise image editing is reasonable, and the proposed method demonstrates its effectiveness on several benchmarks. However, the proposed method in the manuscript only uses human-annotated data, although the semi- or un-supervised segmentation labels should be able to be used to scale up the training. Therefore, I recommend "borderline accept".

**Limitations:**

Yes.

**Paper Formatting Concerns:**

N/A.

**Quality:**

2

**Strengths And Weaknesses:**

- Strengths:

1. The motivation and idea of using predicted segmentation masks for better controlling precise image editing is reasonable.
2. The dual encoders is carefully designed to facilitate catching both global and fine-grained information.
3. The overall performance of the proposed method compared to the provided baselines is good.

- Weaknesses:

1. The model uses a large amount of human-annotated segmentation training data, which is hard to scale-up.
2. Lack of carefully ablation studies of the proposed key components.
3. Lack of discussions and comparisons on existing works which combine LLM and specialized decoders for generating heterogeneous data such as [1-4].

[1] NExT-GPT: Any-to-Any Multimodal LLM, on ICML'24.
[2] VisionLLM: Large Language Model is also an Open-Ended Decoder for Vision-Centric Tasks, on NeurIPS'23.
[3] VisionLLM v2: A Generalist Multimodal Large Language Model for Hundeds of Vision-Language Tasks, on NeurIPS'24.
[4] Emerging Properties in Unified Multimodal Pretraining, technical report (optional, since it is not officially published).

---

> ### Author Rebuttal · Authors · 2025-07-30
>
> Dear Reviewer, Thank you for your review and the insightful questions you've raised about our work. Your concerns have helped us to more clearly position the value of our contributions.
>
> **Regarding Weakness 1 & Question 2 (Dependence on manually annotated segmentation data and OOD performance)**
>
> Our method should be extensible and employs a highly scalable strategy that eliminates the need for manual annotation by utilizing an automated data generation pipeline. This pipeline combines GroundDINO for object localization, SAM2 for precise segmentation, and a Large Vision-Language Model (LVLM) for further refinement, enabling the efficient production of high-quality training data.
>
> Meanwhile, our model (FOCUS) achieves state-of-the-art (SOTA) performance on out-of-distribution (OOD) benchmarks, including ADE-OV, Citys-OV, PC59-OV, and PAS20-OV. This outstanding performance is largely due to the integrated LLM’s advanced language understanding capabilities for visual grounding, which enable robust results even when processing unfamiliar instructions across a wide range of scenarios.
>
> Table 1: Comparison of segmentation performance on out-of-distribution (OOD) benchmarks.
>
> | Method   | ADE-OV |      | Citys-OVs | PC59-OV | PAS20-OV |
> | :------- | :----- | :--- | :-------- | :------ | -------- |
> |          | PQ     | mIoU | PQ        | mIoU    | mIoU     |
> | PSALM    | 13.7   | 18.2 | 28.8      | 48.5    | 81.3     |
> | HyperSeg | 16.1   | 22.3 | 31.1      | 64.6    | 92.1     |
> | FOCUS(Ours)    | 18.5   | 25.7 | 33.4      |  67.3   |  93.8    |
>
> **Regarding Weakness 2 Lack of ablation studies**
>
> *   **The effect of multi-stage training:**
>
>     The following table shows the results of using different image resolutions (256, 512, and 1024) for all-stage task training. We observe a clear trend: higher image resolutions improve the model’s performance in image generation and segmentation, but lead to a decline in language understanding. Our analysis suggests that this decline is mainly due to an increase in hallucination issues at higher resolutions—when processing high-resolution images, the model is more likely to generate text that is inconsistent with the visual input. This increased hallucination negatively impacts the model’s language comprehension, highlighting the challenge of balancing visual precision and robust language understanding in multimodal models.
>
>     To address this issue, our multi-stage training strategy starts with simpler tasks at lower image resolutions and gradually increases both the image size and task complexity in subsequent stages. This progressive approach enables our model to achieve strong and balanced performance across all tasks.
>
>     Table 2: The effect of different image resolutions and multi-stage training strategy on model performance across various tasks. ↑ indicates higher is better, ↓ indicates lower is better.
>     | Image Size | MJHQ30K | GenAI-Bench |         | Image Understanding |         |         | RefCOCO  |          |          |
>     | :--------- | :------ | :---------- | :------ | :------------------ | ------- | ------- | -------- | -------- | -------- |
>     |            | FID(↓)  | Basic(↑)    | Adv.(↑) | POPE(↑)             | MMB(↑)  | SEED(↑) | testA(↑) | TestB(↑) | Valid(↑) |
>     | 256        | 11.85   | 0.63        | 0.56    | 85.3                | 73.1    | 70.1    | 78.5     | 79.2     | 75.4     |
>     | 512        | 7.39    | 0.70        | 0.61    | 87.3                | 80.6    | 71.3    | 81.3     | 82.4     | 79.8     |
>     | 1024       | 6.03    | 0.76        | 0.69    | 77.6(↓)             | 68.2(↓) | 68.8(↓) | 83.4     | 84.9     | 81.5     |
>     | 256->1024(Our)        | 6.05    | 0.83        | 0.72    | 88.0                | 81.5    | 73.9    | 84.1     | 86.3     | 82.7     |
>
> *   **The effect of multi-scale features in Gated Cross-Attention.**
>
>     To effectively fuse multi-scale features, our model employs a Gated Cross-Attention module to process outputs from the ConvNeXt-L backbone, achieving a balance between fine-grained detail and global contextual understanding. Specifically, we design a **3-layer** Gated Cross-Attention adapter that hierarchically integrates three distinct feature scales: the first layer focuses on fine features ($f_2$), the second on mid-level features ($f_3$), and the third on coarse semantic features ($f_4$).
>
>     As illustrated in the table, we compare our three-scale model ($f_2$, $f_3$, $f_4$) with variants that utilize only a single scale ($f_4$), dual scales ($f_3$, $f_4$), and all four scales ($f_2$–$f_4$). The results (mIoU on RefCOCO) demonstrate that our multi-scale fusion strategy is essential for accurately segmenting fine details, providing the optimal balance between performance and computational efficiency.
>
>     Table 3: Ablation study on the effect of multi-scale feature fusion in Gated Cross-Attention.
>
>     | Model Variant              | #Scales | RefCOCO mIoU(↑) | MMBench Acc.(↑) | CLIP-T(Edit)(↑) |
>     | :------------------------- | :------ | :-------------- | :-------------- | :-------------- |
>     | FOCUS-SingleScale (Fine)   | 1       | 79.5            | 72.8            | 0.265           |
>     | FOCUS-SingleScale (Coarse) | 1       | 78.2            | 73.9            | 0.260           |
>     | FOCUS-DualScale            | 2       | 80.5            | 73.5            | 0.270           |
>     | FOCUS (Ours)               | 3       | 81.4            | 73.9            | 0.275           |
>
>
> **Regarding Weakness 3 (Comparison with works like NExT-GPT, VisionLLM, etc.):**
>
> Previous frameworks such as NExT-GPT and VisionLLM v1 & v2 have made progress toward unifying vision tasks, but their reliance primarily on text conditioning limits their ability to achieve precise spatial control in image editing. In contrast, our method stands out for its pixel-aware design: by utilizing the edge information of the target region, our approach enables highly accurate and localized modifications.
>
> This advantage is consistently reflected across multiple benchmarks. **Tables 6 and 7** showcase our superior performance in text-guided video editing (DAVIS) and image editing (COCO-caption), respectively. Furthermore, as illustrated in **Table 8**, our model achieves segmentation accuracy on par with state-of-the-art (SOTA) methods such as VisionLLMv2, while uniquely offering robust editing and generation capabilities.
>
> We also recognize Bagel, a concurrent work with promising results, as an important reference point. We believe that future progress can be achieved by drawing on the strengths and insights of both approaches.
>
> Table 6: Quantitative comparison of text-guided video editing methods on DAVIS.
>
> | Method    | CLIP-T(↑) | CLIP-I(↑) |
> | :-------- | :-------- | :-------- |
> | CogVideo  | 0.2391    | 0.9064    |
> | TuneVideo | 0.2758    | 0.9240    |
> | SDEdit    | 0.2775    | 0.8731    |
> | Pix2Video | 0.2891    | 0.9767    |
> | NExT-GPT  | 0.2684    | 0.9647    |
> | FOCUS(Ours)      | 0.2943    | 0.9815    |
>
>
> Table 7: Quantitative comparison of text-guided image editing methods on COCO-caption.
>
> | Method   | Object  |        | Background |        |
> | :------- | :------ | :----- | :--------- | :----- |
> |          | CLIP(↑) | FID(↓) | CLIP(↑)    | FID(↓) |
> | PTP      | 30.33   | 9.58   | 31.55      | 13.92  |
> | BLDM     | 29.95   | 6.14   | 30.38      | 20.44  |
> | DiffEdit | 29.30   | 3.78   | 26.92      | 1.74   |
> | PFB-Diff | 30.81   | 5.93   | 32.25      | 13.77  |
> | NExT-GPT | 29.32   | 6.62   | 27.31      | 14.27  |
> | FOCUS(Ours)    |  31.22  |  3.45  |  33.10     |  1.98  |
>
> Table 8: Quantitative comparison of segmentation performance on RefCOCO and ReasonSeg benchmarks.
>
> | Method      | RefCOCO |          |          | ReasonSeg |
> | :---------- | :------ | :------- | -------- | --------- |
> |             | Val(↑)  | testA(↑) | testB(↑) | gIoU(↑)   |
> | VisionLLMv2 | 76.6    | 79.3     | 74.3     | 51.0      |
> | PSALM       | 83.6    | 84.7     | 81.6     | -         |
> | HyperSeg    | 84.8    | 85.7     | 83.4     | 59.2      |
> | FOCUS(Ours)       |  84.1   | 86.3     |  82.7    |  61.6     |
>
> **Regarding Question 1 the role of the segmentation mask**
>
> To clearly illustrate the fundamental importance of the segmentation mask in our framework, we have designed a series of comparative experiments:
>
> 1.  **FOCUS (Full Model):** Utilizes internally generated masks that are precisely aligned with the given language instructions.
> 2.  **Baseline 1 (No Mask Guidance):** Removes mask guidance entirely, relying solely on text instructions to guide the diffusion model during editing.
> 3.  **Baseline 2 (General-Purpose Mask):** Employs masks produced by an external, pre-trained, general-purpose segmentation model (Grounding+SAM2) for guidance.
>
> These results demonstrate that precise and instruction-aligned segmentation masks are essential for optimal editing performance in our framework.
>
> Table 9: Ablation study on the effect of segmentation mask guidance on EmuEdit benchmark.
>
> | Method                           | EmuEdit |           |           |             |
> | :------------------------------- | :------ | --------- | --------- | ----------- |
> |                                  | DINO(↑) | CLIP-I(↑) | CLIP-T(↑) | CLIP-DIR(↓) |
> | **No Mask Guidance**             | 0.812   | 0.866     | 0.268     | 0.113       |
> | **Using a General-Purpose Mask** | 0.819   | 0.868     | 0.271     | 0.108       |
> | **FOCUS(Ours)**                  | 0.826   | 0.872     | 0.275     | 0.101       |

---

> > ### Author Response · Authors · 2025-08-04
> >
> > Thank you for reviewing our paper. We have carefully addressed the questions raised in your review. Please let us know if there are any remaining concerns or if all issues have been sufficiently resolved.

---

> > ### Author Response · Authors · 2025-08-07
> >
> > We sincerely appreciate the time you've already invested in reviewing our paper. We've done our best to address the concerns you raised and would be grateful if you could take a look when your schedule permits.
> >
> > We understand how busy this period can be. If you need more time or if there's anything unclear in our response, please let us know.

---

> > ### Comment · Reviewer_9FR4 · 2025-08-09
> >
> > I appreciate the authors for their insightful response. After carefully read the response as well as the reviews and discussions from other reviewers, I believe this paper is acceptful for NeurIPS 2025. Therefore, I raise my score to 4.

---

> > > ### Author Response · Authors · 2025-08-09
> > >
> > > Thank you very much for your thoughtful reconsideration and for raising your score. We greatly appreciate the time and effort you invested in thoroughly reviewing our work and engaging in this constructive discussion. Your initial feedback helped us better articulate the strengths and address the limitations of our approach. We believe the additional experimental results and clarifications have strengthened the paper significantly.

---

> ### Comment · Area_Chair_pJRZ · 2025-08-07
>
> Dear reviewer,
>
> The author has responded to your question. As the discussion period is nearing its end, please review their reply to see if it addresses your concerns or if you have any follow-up questions. Kindly remember to submit your final review by the deadline.
>
> Area Chair

---

### Official Review · Reviewer_WJ6V · 2025-07-05

**Clarity:** 3
**Significance:** 4
**Originality:** 3
**Rating:** 5
**Confidence:** 3

**Summary:**

The paper present FOCUS, a unified vision-language model that tackles the problems of segmentation-aware image understanding (high-level) and image generation (low-level).  The authors propose a novel dual-branch visual encoder composed of two encoders that extract the fine (ConvNext) and coarse-grained (QwenViT) information from the input image. Additionally, FOCUS also utilises a visual tokeniser for image generation, along with continuous tokens, to account for the loss of information that occurs during the quantisation process. The continuous tokens provide fine-grained details to the LLM for image understanding tasks. The LLM outputs the mask embeddings, prompt embeddings, image quantizer embeddings, and the semantic recognition tokens. The mask embeddings and the prompt embeddings are used in the mask decoder module for the referring object segmentation task, while the quantised image embeddings, along with decode masks, are used in the diffusion decoder for image generation and editing tasks. The models are trained in multiple stages (4) to learn referring object segmentation and image generation tasks. FOCUS is evaluated on various tasks (image generation, image understanding, image editing, and referring object segmentation) for multiple datasets, showing performance improvements over existing methods on all these tasks.

**Questions:**

1. How does the dataset size affect the performance of these models? Unified understanding and generation performance. Are the potential improvements due to using larger training data, or does the design decision remain unclear?
2. The authors resize the segmentation masks to the latent spatial resolution. This resolution is much smaller than the full image resolution. How does this affect the editing and generation of objects that are very small in size?
3. Major ablations (see weakness section) are missing. The authors need to quantify the impact of each component of the model architecture and training. For example, what is the impact of the decoding constraints the authors impose when decoding the tokens from the LLM? These would help us understand the impact of the work further

**Ethical Concerns:**

["NO or VERY MINOR ethics concerns only"]

**Final Justification:**

I have read the final responses from the author and their rebuttal. I would be happy to keep my original rating of accept for this paper.

**Limitations:**

The paper does not have a dedicated limitation section (main paper and supplemental), which the authors say they provide in the checklist.
1. One limitation I see is in editing small objects, as the masks are downsampled to the latent spatial resolution.
2. Another limitation I see is that replicating the work is challenging due to the high computational resources required for training the model.
3. Due to reliance on SDXL for editing the models also inherits the limitation of these models, For example, in Figure 5 is the models are able to make edits which change the spatial relations between the objects For example, in image 5: the Prompt “make the cat sit on top of dogs head”

**Paper Formatting Concerns:**

Please see the weakness section (W3).

**Quality:**

3

**Strengths And Weaknesses:**

- Strengths:
    1. The paper is well written. The motivation for the problem the authors are solving and how it differs from other VLM models is explained well.
    2. The methodology is well explained. The authors clearly explain everything, and the details about the method are well presented.
    3. The paper presents impressive results and outperforms many baseline methods in image understanding (across multiple datasets), image generation, referring segmentation, and region-level editing tasks.
    4. The paper introduces a novel dual-encoder module that combines high- and low-level features of images required to solve joint image understanding and generation tasks.
    5. The paper demonstrates good results for temporal consistency in tracking and segmenting objects across the video, as shown in Table 5 of the appendix.

- Weakness:
    1. Missing ablations that show the impact of each change in the paper. For example,
        - What is the effect of multi-stage training, which gradually increases the image sizes and task complexity?
        - What is the effect of increasing or decreasing the number of multi-scale features in the gated cross-attention module? It would be good to see qualitative and quantitative results here.
        - The effect of using the continuous tokens in the visual tokeniser.
        - The effect of putting constraints on the decoding strategy in LLM.
    2. The paper claims to have a dedicated limitations section, discussing the limitations of the work, which I was unable to find.
    3. There are several writing issues in the experiment section:
        - No citation for dataset used in Table 1, eg, POPE, MMBench, etc. All datasets or works that the authors use in their research should be properly cited. The authors have citations for these in the supplementary material, but not in the main paper.
        - Table 2 has incorrect bolding of numbers in multiple cases; similarly, Table 3 lacks bolding altogether.
        - The authors do not provide citations to FID, MJHQ metrics, etc. The authors state in the checklist that all datasets and code used are cited, but this is not the case.
  4. One minor point regarding the writing for me is that the paper places some important details and visualisations (Figs. 1 and 2) in the supplementary section. If these details were in the main paper, it would make understanding the methodology much easier.

---

> ### Author Rebuttal · Authors · 2025-07-30
>
> **Regarding Weakness 1 & Question3 (Lack of ablation studies):**
>
> *   **The effect of multi-stage training:**
>
>     The following table shows the results of using different image resolutions (256, 512, and 1024) for all-stage task training. We observe a clear trend: higher image resolutions improve the model’s performance in image generation and segmentation, but lead to a decline in language understanding. Our analysis suggests that this decline is mainly due to an increase in hallucination issues at higher resolutions—when processing high-resolution images, the model is more likely to generate text that is inconsistent with the visual input. This increased hallucination negatively impacts the model’s language comprehension, highlighting the challenge of balancing visual precision and robust language understanding in multimodal models.
>
>     To address this issue, our multi-stage training strategy starts with simpler tasks at lower image resolutions and gradually increases both the image size and task complexity in subsequent stages. This progressive approach enables our model to achieve strong and balanced performance across all tasks.
>
>     Table 1: The effect of different image resolutions and multi-stage training strategy on model performance across various tasks. ↑ indicates higher is better, ↓ indicates lower is better.
>     | Image Size | MJHQ30K | GenAI-Bench |         | Image Understanding |         |         | RefCOCO  |          |          |
>     | :--------- | :------ | :---------- | :------ | :------------------ | ------- | ------- | -------- | -------- | -------- |
>     |            | FID(↓)  | Basic(↑)    | Adv.(↑) | POPE(↑)             | MMB(↑)  | SEED(↑) | testA(↑) | TestB(↑) | Valid(↑) |
>     | 256        | 11.85   | 0.63        | 0.56    | 85.3                | 73.1    | 70.1    | 78.5     | 79.2     | 75.4     |
>     | 512        | 7.39    | 0.70        | 0.61    | 87.3                | 80.6    | 71.3    | 81.3     | 82.4     | 79.8     |
>     | 1024       | 6.03    | 0.76        | 0.69    | 77.6(↓)             | 68.2(↓) | 68.8(↓) | 83.4     | 84.9     | 81.5     |
>     | 256->1024(Our)        | 6.05    | 0.83        | 0.72    | 88.0                | 81.5    | 73.9    | 84.1     | 86.3     | 82.7     |
>
> *   **The effect of multi-scale features in Gated Cross-Attention.**
>
>     To effectively fuse multi-scale features, our model employs a Gated Cross-Attention module to process outputs from the ConvNeXt-L backbone, achieving a balance between fine-grained detail and global contextual understanding. Specifically, we design a **3-layer** Gated Cross-Attention adapter that hierarchically integrates three distinct feature scales: the first layer focuses on fine features ($f_2$), the second on mid-level features ($f_3$), and the third on coarse semantic features ($f_4$).
>
>     As illustrated in the table, we compare our three-scale model ($f_2$, $f_3$, $f_4$) with variants that utilize only a single scale ($f_4$), dual scales ($f_3$, $f_4$), and all four scales ($f_2$–$f_4$). The results (mIoU on RefCOCO) demonstrate that our multi-scale fusion strategy is essential for accurately segmenting fine details, providing the optimal balance between performance and computational efficiency.
>
>     Table 2: Ablation study on the effect of multi-scale feature fusion in Gated Cross-Attention.
>
>     | Model Variant              | #Scales | RefCOCO mIoU(↑) | MMBench Acc.(↑) | CLIP-T(Edit)(↑) |
>     | :------------------------- | :------ | :-------------- | :-------------- | :-------------- |
>     | FOCUS-SingleScale (Fine)   | 1       | 79.5            | 72.8            | 0.265           |
>     | FOCUS-SingleScale (Coarse) | 1       | 78.2            | 73.9            | 0.260           |
>     | FOCUS-DualScale            | 2       | 80.5            | 73.5            | 0.270           |
>     | FOCUS (Ours)               | 3       | 81.4            | 73.9            | 0.275           |
>
> *   **The effect of continuous visual tokens.** Our ablation study demonstrates that feeding **continuous visual features** directly into the LLM is essential for optimal performance. By bypassing discrete tokenization, this approach effectively reduces **information loss** and retains **fine-grained** visual details. As a result, the LLM receives a richer and more precise visual representation, which significantly enhances its ability to comprehend complex multimodal scenarios and accurately interpret referential instructions.
>
>     Table 3: Ablation study on the effect of continuous visual token input on model performance.
>
>     | Continuous Input | MJHQ30K | GenAI-Bench |         | Image Understanding |        |         |
>     | :--------------- | :------ | :---------- | :------ | ------------------- | ------ | ------- |
>     |                  | FID(↓)  | Basic(↑)    | Adv.(↑) | POPE(↑)             | MMB(↑) | SEED(↑) |
>     | **×**                |  6.85   |  0.68       |  0.65   | 82.1                | 50.4   | 56.0    |
>     | **√**               |  6.05   |  0.83       |  0.72   | 85.3                | 70.9   | 66.6    |
>
> *   **The effect of LLM decoding constraints**: The results in the table demonstrate the effectiveness of our decoding constraint, which requires the model to first identify object categories before generating segmentation masks. This structured decoding process not only enhances semantic consistency and reduces the risk of arbitrary segmentation from ambiguous instructions, but also leads to a significant improvement in final segmentation accuracy.
>
>     Table 4: Ablation study on the effect of LLM decoding constraints on segmentation performance.
>
>     | **constraints** | YT-VIS (mAP) | RefCOCO (cIoU) |
>     | :-------------- | :----------- | :------------- |
>     | **×**               | 50.8         | 83.5           |
>     | **√**               | 54.6         | 84.1           |
>
>
> **Regarding Weakness 2 & Question2 (Lack of Limitations Discussion)**: Thank you for your feedback. We apologize that the 'Limitations' section, though mentioned in our submission checklist, was omitted from the main manuscript. We will add the limitation: **(1)** the limited precision when editing small objects due to latent space downsampling; **(2)** the high computational cost, which challenges replication; and **(3)** inherited weaknesses from SDXL, such as rendering complex spatial relationships, which we frame as a direction for future work.
>
> **Regarding Weakness 3\&4 (Writing Issues in the Experimental Section)**: Thank you for your suggestions and feedback. We will revise the manuscript to further improve the clarity and completeness of the paper. First, we will add citations in the main paper for all datasets (e.g., POPE, MMBench) and evaluation metrics (e.g., FID, MJHQ) to ensure proper attribution. Second, we have corrected the incorrect bolding in Table 2 and added appropriate bolding in Table 3 to accurately highlight the best-performing results. Finally, we will move Figures 1 and 2 from the supplementary material into the main paper to enhance clarity and readability.
>
> **Regarding Question1**:
>
> The progressive training paradigm we adopt, by elevating task complexity and resolution in stages, has at its core the optimization of data utilization strategy and effectiveness, rather than a reliance on the expansion of data scale.

---

> > ### Comment · Reviewer_WJ6V · 2025-08-04
> >
> > Thank you for providing the detailed ablation results; this enhances the completeness of the paper. Please ensure they are included in the final version. After reviewing your rebuttal and the comments from other reviewers, I will uphold my initial recommendation to accept the paper.

---

> > > ### Author Response · Authors · 2025-08-04
> > >
> > > Thank you very much for your feedback and for upholding your recommendation to accept our paper. We greatly appreciate your constructive comments throughout the review process, which have helped us improve the quality and completeness of our work.
> > > We will ensure that all the ablation results discussed in our rebuttal are properly included in the final version of the paper, as you suggested.
> > > Thank you again for your time and valuable insights.

---

### Decision · Program_Chairs · 2025-09-17

**Decision:**

Accept (poster)

**Comment:**

This paper introduces a new model that jointly performs image / video understanding and generation. Specifically, the authors propose an architecture capable of both segmentation and object-based editing, built on top of a VLM. The reviewers acknowledge that the model design is well-motivated and that it demonstrates impressive results both qualitatively and quantitatively, outperforming baseline methods on a wide range of tasks. On the other hand, the reviewers raise concerns regarding the clarity of presentation, the lack of ablation studies, and the highly complex model design, which lacks strong justification. Most of these concerns were adequately addressed in the rebuttal, though one reviewer remains skeptical about the necessity of the model’s complexity. The Area Chair agrees with the reviewers’ assessment and considers this paper acceptable, conditioned on the information introduced and promised in the rebuttal being properly incorporated into the final version.